# Optimal Transport for Domain Adaptation through Gaussian Mixture Models

**Eduardo Fernandes Montesuma**                                           *edumfern@gmail.com*
*Université Paris-Saclay, CEA, List, F-91120 Palaiseau France*

**Fred Ngolè Mboula**                                                *fred.ngole-mboula@cea.fr*
*Université Paris-Saclay, CEA, List, F-91120 Palaiseau France*

**Antoine Souloumiac**                                             *antoine.souloumiac@cea.fr*
*Université Paris-Saclay, CEA, List, F-91120 Palaiseau France*

**Reviewed on OpenReview:** *https://openreview.net/forum?id=DCAeXwLenB*

## Abstract

Machine learning systems operate under the assumption that training and test data are sampled from a fixed probability distribution. However, this assumptions is rarely verified in practice, as the conditions upon which data was acquired are likely to change. In this context, the adaptation of the unsupervised domain requires minimal access to the data of the new conditions for learning models robust to changes in the data distribution. Optimal transport is a theoretically grounded tool for analyzing changes in distribution, especially as it allows the mapping between domains. However, these methods are usually computationally expensive as their complexity scales cubically with the number of samples. In this work, we explore optimal transport between Gaussian Mixture Models (GMMs), which is conveniently written in terms of the components of source and target GMMs. We experiment with 9 benchmarks, with a total of 85 adaptation tasks, showing that our methods are more efficient than previous shallow domain adaptation methods, and they scale well with number of samples $n$ and dimensions $d$.

## 1 Introduction

Supervised machine learning models are trained with significant amounts of labeled data, constituting a training set. The theory of generalization (Redko et al., 2020), provides a theoretical background that guarantees accurate predictions on unseen samples *from the same distribution*. Nonetheless, these models are often forced to predict on related, but different data samples (Quinonero-Candela et al., 2008). This distinction is modeled by a *shift* in the probability distributions generating the data (Sugiyama et al., 2007), which motivates the field of transfer learning (Pan & Yang, 2009), and more specifically the problem of Unsupervised Domain Adaptation (UDA), in which models are adapted from a labeled *source domain*, towards an unlabeled *target domain*, following different distributions[1].

In this context, Optimal Transport (OT) (Villani et al., 2009; Peyré et al., 2019) is a powerful, theoretically grounded framework for comparing and manipulating probability distributions (Montesuma et al., 2024a). This framework works by computing a transportation strategy, that moves one probability distribution into the other at least effort. Based on this core idea, different methods have been proposed for UDA such as Courty et al. (2017); El Hamri et al. (2022); Chuang et al. (2023) and Struckmeier et al. (2023).

---

[1] ⦿  Our code is available at `https://github.com/eddardd/gmm-otda`

However, this methodology faces a few challenges. For instance, OT maps computed between discrete distributions are only defined for samples in the training set. Extrapolating these maps to new samples is the subject of intense research Perrot et al. (2016); Seguy et al. (2017). A possible workaround consists of using Gaussian approximations (Flamary et al., 2019; Struckmeier et al., 2023). While this approach effectively defines a mapping over the whole ambient space, its hypothesis do not reflect the possible sub-populations within the data, which are common in classification problems.

A natural solution to tackle multi-modality in data distributions is using Gaussian Mixture Models (GMMs). A further advantage of this approach is considering the recently proposed GMM OT (GMMOT) by Delon & Desolneux (2020), which establishes an efficient, discrete problem between the components in the GMMs. Furthermore, recent works have established the effectiveness of this idea for *multi-source* domain adaptation (Montesuma et al., 2024b), notably through the use of mixture-Wasserstein barycenters.

Nonetheless, some questions on the use of GMMs for UDA remain open. For instance, Delon & Desolneux (2020) propose different mapping strategies between GMMs, but these either fail to map one GMM into the other, or are subject to randomness when sampling transportation maps between GMM components. Furthermore, if we label the components of GMMs, it is natural to *propagate* these labels, in the sense of Redko et al. (2019) towards the target domain. This paper tackles these two questions.

**Summary of contributions.** In this paper, we propose 2 new strategies for UDA based on GMMs. First, we use basic rules of probability theory for propagating the labels of source domain GMM towards the target domain GMM. We do so, by interpreting the GMMOT plan as the joint probability of source and target GMM components. Second, we map samples from the source domain into the target domain based on the GMMOT plan. For a point in the source domain, our strategy consists of first estimating the component, in the source GMM, most likely to have generated the sample. We then transport this point to components in the target domain, while assigning importance weights based on the GMMOT plan.

**Paper organization.** The rest of this paper is organized as follows. Section 2 presents a few related works on OT for UDA. Section 3 covers the preliminaries on OT and GMMOT theory. Section 4 covers our methodological contributions. Section 5 details our experiments and discussion on UDA. Finally, section 6 concludes this paper.

**Notation.** We use uppercase letters $P$ and $Q$ to denote probability distributions, and $P_S$ and $P_T$ to denote source and target domain distributions. More generally, we use Pr to denote probabilities. For instance, $\Pr(Y = y|X = \mathbf{x})$ denotes the conditional probability of label $Y = y$ given a feature vector $X = \mathbf{x}$. Let $P$ be a distribution over feature vectors. We denote samples from $P$ as $\mathbf{x}^{(P)}$. We reserve $y^{(P)}$ for categorical labels (i.e., $1, \cdots, n_c$, for $n_c$ classes), and $\mathbf{y}^{(P)}$ for its one-hot encoding.

## 2 Related Works

**Optimal transport for domain adaptation.** Optimal transport has been extensively employed for the design of algorithms (Courty et al., 2017), as well as analyzing the domain adaptation problem (Redko et al., 2017). The key idea of this method is to use the Kantorovich formulation to acquire a matching, known as transport plan, between source and target domain distributions. This matching defines a map between points in the source domain, towards the target domain, called barycentric mapping. Based on this idea, different methods have proposed improvements. For instance, Perrot et al. (2016) proposed learning linear and kernelized extensions of the barycentric map through convex optimization. El Hamri et al. (2022) uses clustering for learning matching with additional structural dependencies. Flamary et al. (2019) uses optimal transport between Gaussian distributions for estimating an affine mapping between source and target domains. More recently, Chuang et al. (2023) proposed a method that leverages kernel density estimation for defining a new optimal transport problem based on information maximization.

**Gaussian-mixture based optimal transport.** An optimal transport problem involving Gaussian mixtures was initially proposed by Chen et al. (2018), in which a linear program between the components of the two mixtures is solved. This setting was further studied by Delon & Desolneux (2020), who proved an interesting connection to a continuous optimal transport, when the transport plan is constrained to the set of Gaussian mixtures. Based on the framework of Delon & Desolneux (2020), Montesuma et al. (2024b)

proposed the extension of multi-source domain adaptation algorithms of Montesuma & Mboula (2021a;b) and Montesuma et al. (2023). However, these authors focused on performing adaptation through Wasserstein barycenters. Although based on the same framework, our work focuses on *mapping samples and propagating labels* of Gaussian mixtures, especially for single-source domain adaptation.

## 3 Theoretical Foundations

### 3.1 Optimal Transport

Founded by Monge (1781), optimal transport is a field of mathematics concerned with transporting mass at least effort. Let $\mathcal{X}$ be a set and $\mathcal{P}(\mathcal{X})$ the set of probability distributions on $\mathcal{X}$. For $P, Q \in \mathcal{P}(\mathcal{X})$, the Monge formulation of the optimal transport problem is,

$$T^\star = \operatorname*{arginf}_{T:T_\sharp P=Q} \int_\mathcal{X} c(x, T(x)) dP(x), \tag{1}$$

where $T_\sharp P$ denotes the pushforward distribution (Santambrogio, 2015, Problem 1.1) of $P$ by the map $T$, and $c : \mathcal{X} \times \mathcal{X} \to \mathbb{R}$, called the ground-cost, denotes the cost of sending $x$ to position $T(x)$.

Although equation 1 is a formal description of the optimal transport problem, the constraint $T_\sharp P = Q$ poses technical difficulties. An alternative description was proposed by Kantorovich (1942), in terms of an optimal transport *plan* $\gamma$,

$$\gamma^\star = \operatorname*{arginf}_{\gamma \in \Gamma(P,Q)} \int_{\mathcal{X} \times \mathcal{X}} c(x_1, x_2) d\gamma(x_1, x_2), \tag{2}$$

where $\Gamma(P, Q)$ is the set of joint distributions with marginals $P$ and $Q$. This formulation is simpler to analyze because the constraint $\gamma \in \Gamma(P, Q)$ is linear with respect to the optimization variable $\gamma$.

When $(\mathcal{X}, d)$ is a metric space, it is possible to define a distance on $\mathcal{P}(\mathcal{X})$ in terms of $d$. Let $\alpha \in [1, +\infty)$, and $c(\cdot, \cdot) = d(\cdot, \cdot)^\alpha$. One then has the so-called $\alpha-$Wasserstein distance:

$$\mathcal{W}_\alpha(P,Q)^\alpha = \inf_{\gamma \in \Gamma(P,Q)} \int_{\mathcal{X} \times \mathcal{X}} c(x_1, x_2) d\gamma(x_1, x_2). \tag{3}$$

In the following, we do optimal transport on Euclidean spaces, i.e., $\mathcal{X} = \mathbb{R}^d$. In this case, it is natural to use $d(\mathbf{x}_1, \mathbf{x}_2) = \|\mathbf{x}_1 - \mathbf{x}_2\|_2$. Furthermore, we set $\alpha = 2$. Equations 2 and 3 are linear programs, where the optimization variable is the joint distribution $\gamma$. In the following, we discuss 3 particular cases where optimal transport either has a closed for, or is approximated by a finite problem, thus tractable by a computer.

**Empirical Case.** If we have samples $\{\mathbf{x}_i^{(P)}\}_{i=1}^n$ and $\{\mathbf{x}_j^{(Q)}\}_{j=1}^m$ with probabilities $p_i$ and $q_j$ respectively, we can make empirical approximations for $P$ and $Q$,

$$\hat{P}(\mathbf{x}) = \sum_{i=1}^n p_i \delta(\mathbf{x} - \mathbf{x}_i^{(P)}). \tag{4}$$

The approximation in equation 4 is at the core of discrete optimal transport Peyré et al. (2019). If we plug equation 4 into equation 2, the optimal transport problem becomes computable, i.e., it turns into a linear programming problem with $n \times m$ variables,

$$\gamma^\star = \arg \min_{\gamma \in \Gamma(\hat{P}, \hat{Q})} \sum_{i=1}^n \sum_{j=1}^m \gamma_{ij} C_{ij}, \tag{5}$$

where $C_{ij} = c(\mathbf{x}_i^{(P)}, \mathbf{x}_j^{(Q)})$. As a linear program, one should keep in mind that solving equation 5 has a complexity of $\mathcal{O}(n^3 \log n)$. This complexity can be reduced by regularizing the problem in terms of the OT plan entropy,

$$\gamma^\star = \arg \min_{\gamma \in \Gamma(\hat{P}, \hat{Q})} \sum_{i=1}^n \sum_{j=1}^m \gamma_{ij} C_{ij} + \epsilon \sum_{i=1}^n \sum_{j=1}^m \gamma_{ij} (\log \gamma_{ij} - 1), \tag{6}$$

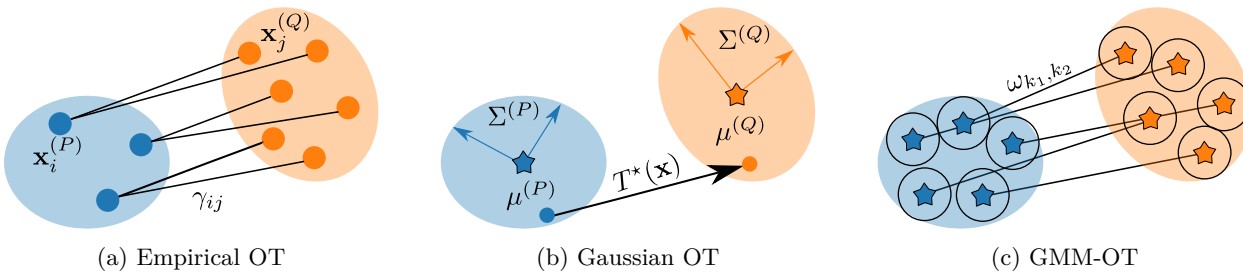

(a) Empirical OT        (b) Gaussian OT        (c) GMM-OT

Figure 1: **Comparison of different ways of solving OT.** In empirical OT, $P$ and $Q$ are approximated non-parametrically through their samples. In Gaussian OT, $P$ and $Q$ are Gaussian distributions, and the mapping between these distributions is affine. In GMM-OT, $P$ and $Q$ are assumed to be GMMs, and an OT plan between components, $\omega$, defines an OT plan between samples, $\gamma$.

which can be solved through the celebrated Sinkhorn algorithm (Cuturi, 2013). A solution can be found with complexity $\mathcal{O}(Ln^2)$, where $L$ is the number of iterations of the algorithm. From the Kantorovich formulation, we can recover a correspondence between distributions through the barycentric map,

$$T_\gamma(\mathbf{x}_i^{(P)}) = \min_{\mathbf{x} \in \mathcal{X}} \sum_{j=1}^{m} \gamma_{ij} c(\mathbf{x}, \mathbf{x}_j^{(Q)}). \tag{7}$$

**Gaussian Case.** When $P = \mathcal{N}(\mu^{(P)}, \Sigma^{(P)})$ (resp., $Q$), equation 1 has a closed and affine form Takatsu (2011), $T^\star(\mathbf{x}) = \mathbf{A}\mathbf{x} + \mathbf{b}$, where,

$$\mathbf{A} = (\Sigma^{(P)})^{-\frac{1}{2}}((\Sigma^{(P)})^{\frac{1}{2}}\Sigma^{(Q)}(\Sigma^{(P)})^{\frac{1}{2}})^{\frac{1}{2}}(\Sigma^{(P)})^{-\frac{1}{2}}, \text{ and } \mathbf{b} = \mu^{(Q)} - \mathbf{A}\mu^{(P)}, \tag{8}$$

and the Wasserstein distance takes the form,

$$\mathcal{W}_2(P,Q)^2 = \|\mu^{(P)} - \mu^{(Q)}\|_2^2 + \mathrm{Tr}(\Sigma^{(P)} + \Sigma^{(Q)} + ((\Sigma^{(P)})^{1/2}\Sigma^{(Q)}(\Sigma^{(P)})^{1/2})^{1/2}). \tag{9}$$

In contrast with empirical optimal transport, under the Gaussian hypothesis, computing the Wasserstein distance and a mapping between $P$ and $Q$ has sample-free complexity. Indeed, the complexity of equations 8 and 9 is dominated by computing the square-root of the covariance matrix with complexity $\mathcal{O}(d^3)$.

Furthermore, in high dimensions and when a only a few data points are available, estimating full covariance matrices is challenging. In these cases, it is useful to assume axis-aligned Gaussians, i.e., $\Sigma$ is a diagonal matrix with diagonal elements $\Sigma_{ii} = \sigma_i^2$. In this case, equations 8 and 9 can be further simplified,

$$\mathbf{A} = \mathrm{diag}(\sigma^{(Q)}/\sigma^{(P)}), \mathbf{b} = \mu^{(Q)} - \mathbf{A}\mu^{(P)}, \text{ and } \mathcal{W}_2(P,Q)^2 = \|\mu^{(P)} - \mu^{(Q)}\|_2^2 + \|\sigma^{(P)} - \sigma^{(Q)}\|_2^2. \tag{10}$$

**Gaussian Mixture Case.** A Gaussian mixture corresponds to $P = \sum_{k=1}^{K} \pi_k^{(P)} P_k$, where $P_k = \mathcal{N}(\mu_k^{(P)}, \Sigma_k^{(P)})$. As in Delon & Desolneux (2020), we denote by $\mathrm{GMM}_d(K)$ the set of distributions $P \in \mathcal{P}(\mathbb{R}^d)$ written as a mixture of at most $K$ components. In this framework Delon & Desolneux (2020) explores the optimal transport problem 2 under the constraint that $\gamma$ is a GMM as well, i.e., $\gamma \in \Gamma(P,Q) \cap GMM_{2d}(\infty)$. This formulation is interesting because it is equivalent to a discrete and hierarchical problem (Delon & Desolneux, 2020, Proposition 4) in terms of the GMMs' components

$$\omega^\star = \mathrm{GMMOT}(P,Q) = \underset{\omega \in \Gamma(\pi^{(P)}, \pi^{(Q)})}{\arg\min} \sum_{k_1=1}^{K_P} \sum_{k_2=1}^{K_Q} \omega_{k_1,k_2} \mathcal{W}_2(P_{k_1}, Q_{k_2})^2. \tag{11}$$

We call this problem GMMOT. This latter equation is an hierarchical optimal transport problem, i.e., a problem that involves itself an inner optimal transport. As in the previous cases, we have a notion of distance related to equation 11, $\mathcal{MW}_2(P,Q)^2 = \sum_{k_1=1}^{K_1} \sum_{k_2=1}^{K_2} \omega_{k_1,k_2} \mathcal{W}_2(P_{k_1}, Q_{k_2})^2$. We show an overview of different strategies for solving OT, under different assumptions, in Figure 1.

**Remark 3.1.** *(Computational Complexity) The overall complexity of the GMMOT in equation 11 is $\mathcal{O}(K^3 \log K)$. However, one should keep in mind that the ground-cost matrix must be computed beforehand, as given by equation 9, which involves a complexity that scales with the dimension of the ambient space, due the matrix inversions and square-roots. The complexity of these operations is $\mathcal{O}(K^2 d^3)$ in general. However, assuming diagonal covariance matrices, the computational complexity is drastically reduced to $\mathcal{O}(K^2 d)$, i.e., the complexity of computing $K^2$ Euclidean distances between $d-$dimensional vectors. We refer readers to our appendix for a running time analysis of our method.*

**Remark 3.2.** *(Parameter Estimation) Besides the computational advantage, using diagonal covariance matrices yields a simpler estimation problem for GMMs. This is pivotal in Domain Adaptation (DA), as the target domain likely does not have enough samples for the accurate estimation of complete covariance matrices. Furthermore, data is oftentimes high-dimensional, as feature spaces commonly involve thousands of features. This is a sharp contrast with previous works in GMMOT, such as Delon & Desolneux (2020) and Chen et al. (2018), which involved a few dimensions. In our empirical validation (c.f., section 5.3) we show that using diagonal covariances yields better adaptation performance in high dimensions.*

### 3.2 Learning Theory and Domain Adaptation

In this paper, we deal with DA for classification. This latter problem can be formalized mathematically, as the learning of a function $h : \mathcal{X} \to \mathcal{Y}$, from a feature space $\mathcal{X}$ (e.g., $\mathbb{R}^d$) to a label space, $\mathcal{Y} = \{1, \cdots, n_c\}$ through samples of a probability distribution. As reviewed by Redko et al. (2020), from the point of view of probability, there multiple ways of formalizing this problem. Here, we use the Empirical Risk Minimization (ERM) framework of Vapnik (2013). For a probability distribution $P$, a loss function $\mathcal{L} : \mathcal{Y} \times \mathcal{Y} \to \mathbb{R}_+$, and a family of classifiers $\mathcal{H}$, one may define a notion of disagreement between pairs $h, h' \in \mathcal{H}$,

$$\mathcal{R}_P(h, h') = \mathbb{E}_{\mathbf{x} \sim P}[\mathcal{L}(h(\mathbf{x}), h'(\mathbf{x}))]. \tag{12}$$

Equation 12 defines the risk of $h$ with respect $h'$. Given a ground-truth labeling function $h_0 : \mathcal{X} \to \mathcal{Y}$, classification can be phrased in terms of minimizing the risk of $h$ with respect the ground-truth $h_0$, i.e.,

$$h^\star = \arg\min_{h \in \mathcal{H}} \mathcal{R}_P(h, h_0).$$

Henceforth, we adopt $\mathcal{R}_P(h) = \mathcal{R}_P(h, h_0)$ in short. This formalization equates the problem of learning a classifier with an optimization problem. Nonetheless, in practice one does not have access to *a priori* knowledge from $P$ nor $h_0$. In a more realistic scenario, one has samples $\{\mathbf{x}_i^{(P)}, y_i^{(P)}\}_{i=1}^n$, where $\mathbf{x}_i^{(P)} \overset{iid}{\sim} P$, and $y_i^{(P)} = h_0(\mathbf{x}_i^{(P)})$. Based on these samples, one may estimate the risk empirically, by resorting to the approximation in equation 4,

$$\hat{\mathcal{R}}_P(h) = \frac{1}{n} \sum_{i=1}^n \mathcal{L}(h(\mathbf{x}_i^{(P)}), y_i^{(P)}), \text{ and } \hat{h} = \arg\min_{h \in \mathcal{H}} \hat{\mathcal{R}}_P(h). \tag{13}$$

In machine learning literature, the true risk $\mathcal{R}_P(h)$ is called *generalization error*, i.e., the error that a classifier $h$ makes on samples from the distribution $P$. In contrast, algorithms usually minimize the *training error*, $\hat{\mathcal{R}}_P(h)$, i.e., the error of $h$ on the particular examples available during training.

A key limitation of the presented theory is its assumption that data originates from a single probability distribution $P$. As discussed by Quinonero-Candela et al. (2008), this is seldom happens in practice. For instance, in fault diagnosis, process conditions influence the statistical properties of measured signals (Montesuma et al., 2022). As a result, generalization must be carried to a new, related probability distribution. This problem is known in the literature as DA, a sub-field within transfer learning.

As discussed by Pan & Yang (2009), in transfer learning, a domain is a pair $\mathcal{D} = (\mathcal{X}, P)$ of a feature space and a probability distribution over $\mathcal{X}$. Likewise, a task is a pair $\mathcal{T} = (\mathcal{Y}, h_0)$ of a label space and a ground-truth labeling function. Transfer learning is characterized by different source and target domains and tasks, i.e., $(\mathcal{D}_S, \mathcal{T}_S, \mathcal{D}_T, \mathcal{T}_T)$, where at least one element from the source is different from the target. In this work, we assume different domains $\mathcal{D}_S \neq \mathcal{D}_T$ but the same label space, $\mathcal{Y}_S = \mathcal{Y}_T = \{1, \cdots, n_c\}$.

In this paper, we deal primarily with *distributional shift*. In this case, we assume $\mathcal{X}_S = \mathcal{X}_T = \mathbb{R}^d$, so that source and target domains are characterized by different probability distributions $P_S \neq P_T$. Furthermore, we place ourselves in the *unsupervised DA* setting, that is, we assume labeled source domain data, $\{\mathbf{x}_i^{(P_S)}, y_i^{(P_S)}\}_{i=1}^n$, and unlabeled target domain data $\{\mathbf{x}_j^{(P_T)}\}_{j=1}^m$. Our goal is to use these samples to learn a classifier $h$ that works well on the target domain, i.e., that achieves small target risk $\mathcal{R}_{P_T}$.

From the point of view of DA, the quality and similarity of source domain data plays a prominent role, since supervision comes from this distribution. For instance, if source domain data contains noisy labels, these can be transferred to the target domain, leading to poor results. Likewise, the success of domain adaptation is correlated with the distance, in distribution, between these two domains. We refer readers to (Ben-David et al., 2010; Redko et al., 2017) for further discussion.

**Optimal Transport for Domain Adaptation** was proposed by Courty et al. (2017), and primarily tries to match samples from $P$ to those of $Q$ based on the empirical OT problem in equation 5. After acquiring $\gamma^\star$, the authors propose mapping samples from $P$ towards those of $Q$ via the baryncetric map. This strategy effectively constitutes a new dataset $\{T_{\gamma^\star}(\mathbf{x}_i^{(P)}), y_i^{(P)}\}_{i=1}^n$, where $T_{\gamma^\star}$ is defined by equation 7. Note, here, that under $T_{\gamma^\star}$, the source domain points *carry their labels*, to the target domain. This operation is valid, as long as the conditionals $P_S(Y|X) = P_T(Y|T(X))$, which is restrictive, but reasonable under the covariate shift hypothesis. A few problems plague this strategy. First, $T_{\gamma^\star}$ is only defined on the support of $P$. The mapping of new points has been extensively studied in the literature (Perrot et al., 2016; Seguy et al., 2017; Chuang et al., 2023). Second, it is desirable to have mappings with additional structure with respect the classes in DA. This problem is partially solved by considering special regularization schemes, as covered in Courty et al. (2017). Third, this method is not scalable with respect the number of samples $n$, due its prohibitive complexity $\mathcal{O}(n^3 \log n)$. In this paper, we offer a solution for the aforementioned problems through the GMM-OT framework of Delon & Desolneux (2020).

# 4 Domain Adaptation via Optimal Transport between Gaussian Mixtures

---

**Algorithm 1:** Fitting procedure for GMMs.

1 **function** EM($\mathbf{X}^{(P)}, K_P$)
2    **for** $it = 1, \cdots, n_{iter}$ **do**
      // Expectation Step
3       $G_{i,k} = \dfrac{\pi_k^{(P)} \mathcal{N}(\mathbf{x}_i^{(P)}|\mu_k^{(P)}, \Sigma_k^{(P)})}{\sum_{k'} \pi_{k'}^{(P)} \mathcal{N}(\mathbf{x}_i^{(P)}|\mu_{k'}^{(P)}, \Sigma_{k'}^{(P)})}$;
      // Maximization Step
4       $n_k = \sum_{i=1}^n G_{i,k}$;
5       $\mu_k^{(P)} \leftarrow \dfrac{1}{n_k} \sum_{i=1}^n G_{ik} \mathbf{x}_i^{(P)}$;
6       $\Sigma_k^{(P)} \leftarrow \dfrac{1}{n_k} \sum_{i=1}^n G_{i,k} (\mathbf{x}_i^{(P)} - \mu_k^{(P)})(\mathbf{x}_i^{(P)} - \mu_k^{(P)})^T$;
7       $\pi_k^{(P)} \leftarrow \dfrac{n_k}{n}$;
8    **return** $\mu^{(P)}, \Sigma^{(P)}, \pi^{(P)}$;

---

**Algorithm 2:** Fitting procedure for labeled GMMs.

1 **function** ConditionalEM($\mathbf{X}^{(P)}, \mathbf{Y}^{(P)}, K_P$)
2    $cpc \leftarrow K_P / n_c$;
3    **for** $y = 1, \cdots, n_c$ **do**
      // Samples from y-th class
4       $\mathbf{X}^{(P_y)} \leftarrow \{\mathbf{x}_i^{(P)} : y_i^{(P)} = y\}$;
      // EM on conditionals
5       $\mu^{(P_y)}, \Sigma^{(P_y)}, \pi^{(P_y)} \leftarrow$ EM($\mathbf{X}^{(P_y)}, cpc$);
6       $\nu^{(P_y)} \leftarrow \{\text{one\_hot}(y)\}_{k=1}^{cpc}$;
   // Concatenates all parameters
7    $\mu^{(P)} = \{\{\mu_k^{(P_y)}\}_{k=1}^{cpc}\}_{y=1}^{n_c}$;
8    $\nu^{(P)} = \{\{\nu_k^{(P_y)}\}_{k=1}^{cpc}\}_{y=1}^{n_c}$;
9    $\Sigma^{(P)} = \{\{\Sigma_k^{(P_y)}\}_{k=1}^{cpc}\}_{y=1}^{n_c}$;
10    $\pi^{(P)} = \left\{\left\{\dfrac{\pi_k^{(P_y)}}{\sum_{k=1}^{cpc} \sum_{y=1}^{n_c} \pi_k^{(P_y)}}\right\}_{k=1}^{cpc}\right\}_{y=1}^{n_c}$;
11    **return** $\mu^{(P)}, \Sigma^{(P)}, \nu^{(P)}, \pi^{(P)}$;

---

In this section, we develop new tools for DA through OT between GMMs. As we discussed in our preliminaries section, we are particularly interested in DA for classification. In this context, data is naturally multi-modal, which justifies the mixture modeling. As we previously defined in Section 3, a GMM is a mixture model

with parameters $\{\mu_k^{(P)}, \Sigma_k^{(P)}, \pi_k^{(P)}\}_{k=1}^K$. These parameters can be determined through maximum likelihood:

$$\{\mu_k^{(P)}, \Sigma_k^{(P)}, \pi_k^{(P)}\}_{k=1}^K = \underset{\{\mu_k, \Sigma_k, \pi_k\}_{k=1}^K}{\arg\max} \sum_{i=1}^n \log P(\mathbf{x}_i^{(P)}), \tag{14}$$

where $P = \sum_{k=1}^K \pi_k^K P_k$. A practical approach for optimizing equation 14 was proposed by Dempster et al. (1977), and is known as Expectation Maximization (EM). We show a pseudo-code for this strategy in Algorithm 1. We refer readers to (Bishop M., 2006, Chapter 9) for further details on GMMs.

In our approach, we need to define labels for the components of GMMs in the transportation problem. We do so through an heuristic, that is, we model each $P_y = P(X|Y = y)$ through a GMM, for $y = 1, \cdots, n_c$. As a result, we fit a GMM to the data $\mathbf{X}^{(P_y)} = \{\mathbf{x}_i^{(P)}\}_{i:y_i^{(P)}=y}$, using $cpc = K_P/n_c$ components. Here, we conveniently choose $K_P$ as a multiple of $n_c$, for ensuring that $cpc$ is an integer. We present in Algorithm 2 a pseudo-code for this strategy. Furthermore, we create a one-hot encoded vector of component labels, denoted $\nu^{(P)} \in (\Delta_{n_c})^K$, where $\nu_{k,y}^{(P)} = 1$ if the $k-$th component comes from the $y-$th class, and 0 otherwise. We interpret the vector $\nu_k^{(P)}$ as the conditional probability $\Pr(Y|K = k)$.

In the following, we discuss two strategies for domain adaptation. The first, based on label propagation, leverages the optimal transport plan *between components* to define pseudo-labels for the components of the target domain GMM. The second, based on mapping estimation, leverages the hierarchical nature of the GMMOT problem for defining a map between source and target domain. These methods are summarized in Algorithms 3 and 4.

---

**Algorithm 3:** Pseudo-label target GMM.

1 **function**
   **propagate_labels($\mathbf{X}^{(P_S)}, \mathbf{Y}^{(P_S)}, \mathbf{X}^{(P_T)}$)**
2     $P \leftarrow$ CondtionalEM($\mathbf{X}^{(P_S)}, \mathbf{Y}^{(P_S)}$);
3     $Q \leftarrow$ EM($\mathbf{X}^{(P_T)}$);
4     $\omega \leftarrow$ GMMOT($P_S, P_T$);
5     $\nu^{(P_T)} = \omega^T \nu^{(P_S)}/\pi^{(P_T)}$;
6     **return** $\nu^{(P_T)}$;

---

**Algorithm 4:** $T_{weight}$.

1 **function** $T_{weight}(\mathbf{x}^{(P_S)}, y^{(P_S)}, P_S, \omega, \tau)$
      // Using Equation 17
2     $k_1 \leftarrow$ estimate_components($\mathbf{x}^{(P_S)}, P_S$);
3     **for** $j$ such that $\omega_{k_1,j} \geq \tau$ **do**
        // Using eq. 8 or 10
4       $\tilde{\mathbf{x}}_{k_2}^{(P_S)} \leftarrow T_{k_1,k_2}(\mathbf{x}^{(P_S)})$;
5       $\tilde{y}_{k_2}^{(P_S)} \leftarrow y^{(P_S)}$
6       $w_{k_2} \leftarrow \omega_{k_1,k_2}$
7     **return** $\{w_{k_2}, \tilde{\mathbf{x}}_{k_2}^{(P)}, y_{k_2}^{(P)}\}_{k_2:\omega_{k_1,k_2}\geq\tau}$;

---

**Limitations.** In this work, we assume that data is multi-modal, which is often the case in classification. Furthermore, we assume that it can be modeled accurately through GMMs, i.e., data is well separated into clusters. While we can expect estimation and inference to be difficult in high-dimensions, we show in our experiments that our method outperform previous baselines based on empirical OT. Finally, we show in our appendix that our methods are robust to the *over estimation* of GMM components.

## 4.1 Label Propagation and Maximum a Posterior Estimation

Recall that, in equation 11, the result of the GMMOT problem is a transportation plan $\omega$, between components of the GMMs $P$ and $Q$. As a result, $\omega$ has as marginals the probability vectors $\pi^{(P)}$ and $\pi^{(Q)}$. Furthermore, in the original probabilistic view of GMMs, $\pi_{k_1}^{(P_S)} = \Pr(K_S = k_1)$, whereas $\pi_{k_2}^{(P_T)} = \Pr(K_T = k_2)$. Given this interpretation, it is natural to see the transportation plan as $\omega_{k_1,k_2} = \Pr(K_S = k_1, K_T = k_2)$. Note that we can estimate the probability $\Pr(Y|K_T)$ using the law of total probabilities,

$$\Pr(Y|K_T = k_2) = \sum_{k_1=1}^{K_S} \Pr(Y|K_S = k_1, K_T = k_2)\Pr(K_S = k_1|K_T = k_2).$$

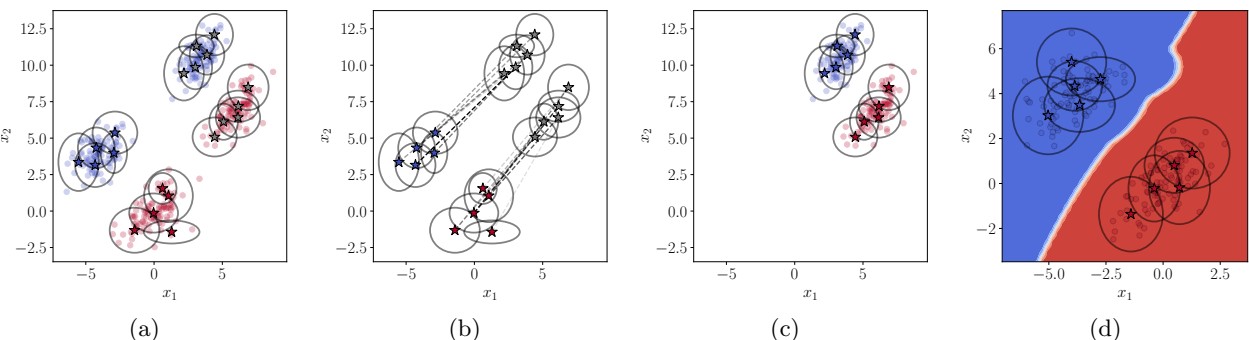

(a)         (b)         (c)         (d)

Figure 2: **Illustration of the label propagation strategy.** (a) Shows the source and target GMMs. GMMs are represented through their means (stars) and covariance matrices (ellipses). The labels of components are represented through colors. Since the target domain is unlabeled, their means are gray colored. (b) Shows the OT plan between components, $\omega$. $\omega$ allows us to propagate the labels of source domain GMM towards the target. (c) Shows the obtained target GMM. Finally, (d) shows the MAP classifier.

Here, assuming that $Y$ and $K_2$ are conditionally independent given $K_1$, we have,

$$\Pr(Y|K_T = k_2) = \sum_{k_1=1}^{K_S} \Pr(Y|K_S = k_1)\Pr(K_S = k_1|K_T = k_2),$$

This assumption plays the same role as covariate shift hypothesis (Sugiyama et al., 2007) in conventional DA works. On an intuitive level, our assumption explicit the fact that $K_T$ is redundant with respect to $K_S$. The conditional $\Pr(K_S|K_T) = \Pr(K_S, K_T)/\Pr(K_T)$ can be computed through the optimal transport plan, i.e.,

$$\hat{\nu}_{k_2}^{(P_T)} = \frac{1}{\pi_{k_2}^{(P_T)}} \sum_{k_1=1}^{K_S} \omega_{k_1,k_2} \nu_{k_1}^{(P_S)}, \text{ or, } \hat{\nu}^{(P_T)} = \frac{\omega^T \nu^{(P_S)}}{\pi^{(Q_T)}}, \tag{15}$$

where the division should be understood elementwise. Equation 15 is known in the OT literature as *label propagation* (Redko et al., 2019), and, as we discussed in the related works section, has been used extensively in the context of empirical OT. Given the estimated labels $\hat{\nu}_{k_2}^{(P_T)}$, we effectively defined a labeled GMM for the target domain. Based on this GMM, we can perform Maximum A Posteriori (MAP) estimation to define a classifier in the target domain,

$$\hat{h}_{MAP}(\mathbf{x}) = \arg\max_{y=1,\cdots,n_c} \Pr(Y = y|X = \mathbf{x}) = \arg\max_{y=1,\cdots,n_c} \sum_{k=1}^{K_T} \Pr(Y = y|X = \mathbf{x}, K_T = k)\Pr(K_T = k|X = \mathbf{x}),$$

$$= \arg\max_{y=1,\cdots,n_c} \sum_{k=1}^{K_T} \left( \frac{\pi_k^{(P_T)} P_{T,k}(\mathbf{x})}{\sum_{k'=1}^{K_T} \pi_{k'}^{(P_T)} P_{T,k'}(\mathbf{x})} \right) \nu_k^{(P_T)}, \tag{16}$$

where, from the second to the third equality we assumed that $Y$ and $K$ are conditionally independent given $X$. This hypothesis is intuitive, as classes and components are representing the same structures within the data points. We show an illustration of these ideas in Figure 2.

## 4.2 Mapping Estimation

In this section, we propose a new mapping estimation technique between two GMMs, $P_S$ and $P_T$. As discussed in (Delon & Desolneux, 2020, Section 6.3), this problem is not straightforward. Indeed, since $\gamma$ is a GMM, it cannot be, in general, written as $(Id, T)_\sharp P_S$. Thus, these authors consider two strategies: a mean map and a random map. First, samples are mapped $T_{mean}(\mathbf{x}^{(P_S)}) = \mathbb{E}_{\mathbf{x}^{(P_T)} \sim \gamma(\cdot|\mathbf{x}^{(P_S)})}[\mathbf{x}^{(P_T)}]$, but it may end up not actually matching $P_S$ with $P_T$ (see, e.g., (Delon & Desolneux, 2020, Section 6.3)).

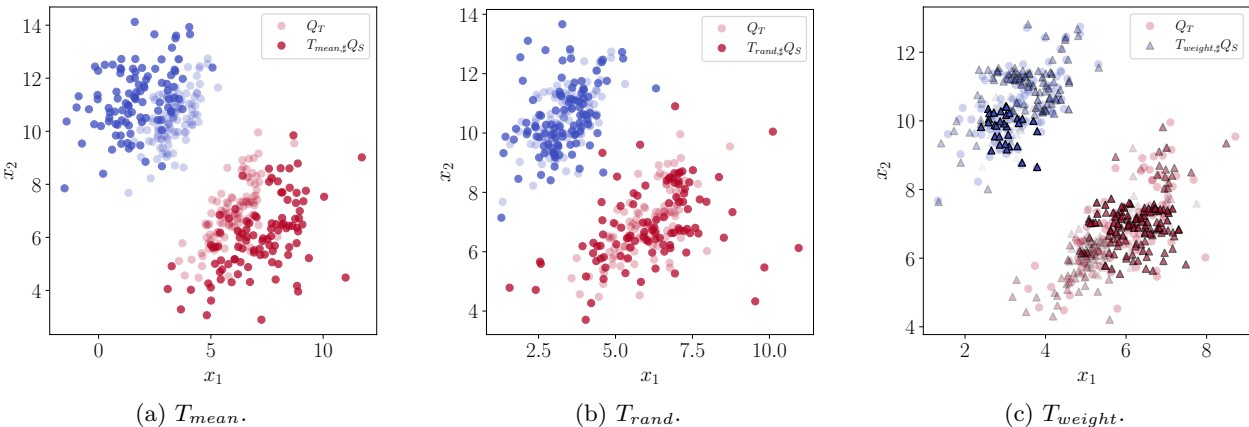

(a) $T_{mean}$.      (b) $T_{rand}$.      (c) $T_{weight}$.

Figure 3: **Mapping estimation using GMMOT.** In (a) and (b), we show the $T_{mean}$ and $T_{rand}$ strategies of Delon & Desolneux (2020), respectively. In (c), we show our strategy $T_{weight}$. Our mapping reduces randomness by first estimating the component $k_1$ most likely to have generated $\mathbf{x}^{(P)}$. Then, we weight the importance of transported samples by $\omega_{k_1,k_2}$.

Second, Delon & Desolneux (2020) defines

$$T_{rand}(\mathbf{x}^{(P_S)}) = T_{k_1,k_2}(\mathbf{x}^{(P_S)}) \text{ with probability } p_{k_1,k_2}(\mathbf{x}^{(P_S)}) = \omega^{\star}_{k_1,k_2} \frac{\mathcal{N}(\mathbf{x}^{(P_S)}|\mu^{(P_S)}_{k_1}, \Sigma^{(P_S)}_{k_1})}{\sum_k p_k \mathcal{N}(\mathbf{x}^{(P_S)}|\mu^{(P_S)}_k, \Sigma^{(P_S)}_k)},$$

which has the advantage of matching $P_S$ with $P_T$. However, as noted by Delon & Desolneux (2020), $T_{rand}$ usually leads to irregular mappings due the sampling procedure of indices $(k_1, k_2)$ as shown in (Delon & Desolneux, 2020, Figure 7).

We put forth a third strategy for mapping $P_S$ into $P_T$. Our intuition is twofold. First, we can increase the regularity of $T_{rand}$, by estimating the component $k_1$ that most likely originated $\mathbf{x}^{(P_S)}$, that is,

$$k_1 := \underset{k=1,\cdots,K_S}{\arg\max} \Pr(K_S = k | X_S = \mathbf{x}^{(P_S)}) = \frac{\pi^{(P_S)}_k P_{S,k}(\mathbf{x}^{(P_S)})}{\sum_{k'=1}^{K_S} \pi^{(P_S)}_{k'} P_{S,k'}(\mathbf{x}^{(P_S)})}. \tag{17}$$

Second, we map $\mathbf{x}^{(P_S)}$ into the components of $P_T$. Note that, since the marginals $\pi^{(P_S)}$ and $\pi^{(P_T)}$ are different, the optimal transport plan $\omega$ may split the mass of $P_{S,k_1}$ into several $P_{T,k_2}$. As a result, we produce $\{T_{k_1,k_2}(\mathbf{x}^{(P_S)})\}_{k_2 : \omega_{k_1,k_2} \geq \tau}$, i.e., we map $\mathbf{x}^{(P_S)}$ to all components $P_{T,k_2}$ such that $\omega_{k_1,k_2} \geq \tau \geq 0$. In principle, one may choose $\tau = 0$ and filter only the components that are not matched with $P_{S,k_1}$. Third, we further weight the importance of generated samples, by using $\omega_{k_1,k_2}$ as sample weights. At the end, we generate a weighted dataset $\{(\omega_{k_1,k_2}, T_{k_1,k_2}(\mathbf{x}^{(P_S)}_i), y^{(P_S)}_i)\}_{i=1}^m$, where $m$ is the total amount of samples generated. We call our overall mapping $T_{weight}$, for which a pseudo-code is presented in 4.

The mapping we just defined has a few interesting properties. First, it is a piece-wise affine map, as each $T_{k_1,k_2}$ is affine. This property contrast with the Gaussian hypothesis, which defines an affine map between $P_S$ and $P_T$. Second, with respect the transportation of samples, our mapping strategy is naturally *group-sparse*, in the sense of Courty et al. (2017). This claim comes from the fact that samples in $P_S$ are transported based on the Gaussian component they belong to. Third, our mapping is defined on the whole support of the GMM $P_S$. In contrast, empirical OT is only defined on the samples $\{\mathbf{x}^{(P_S)}_i\}_{i=1}^n$ of $P_S$.

## 5 Experiments

In this section, we present our experiments with DA. We consider a wide range of 9 benchmarks in computer vision and fault diagnosis. In the first case, we consider Caltech-Office (Gong et al., 2012), ImageCLEF (Caputo et al., 2014), Office31 (Saenko et al., 2010), Office-Home (Venkateswara et al., 2017), (MNIST, USPS,

SVHN) (Seguy et al., 2017) and VisDA Peng et al. (2017). In the second case, we consider the CWRU[2], CSTR (Pilario & Cao, 2017; Montesuma et al., 2022) and TE process benchmarks (Montesuma et al., 2024c). Further details on these benchmarks are available in the appendix. In our experiments, an adaptation task is a pair $(S, T)$ of a source domain $S$ and a target domain $T$. To summarize our experimentation, there are in total 9 benchmarks, and 85 domain adaptation tasks. Our experiments with Cross-Domain Fault Diagnosis (CDFD) are available in our appendix.

For computer vision benchmarks, we follow previous research (El Hamri et al., 2022; Chuang et al., 2023) and pre-train ResNet (He et al., 2016) networks on the source domains. We then use the encoder branch as a feature extractor, and perform *shallow DA* on the extracted features. These feature serve as the basis for each domain adaptation algorithm. With the exception of GMM-OTDA$_{MAP}$, performance on the target domain is based on the generalization of a 1-layer neural network trained with transformed data. For GMM-OTDA$_{MAP}$ we use the MAP strategy described in equation (16). For the *MNIST, USPS, SVHN* benchmark, we follow Seguy et al. (2017) to obtain comparable results. For $M \rightarrow U$ and $U \rightarrow M$ we downsize MNIST to the resolution of USPS, i.e., we downscale images to $(16, 16)$. For $M \rightarrow S$, we upscale MNIST images to match the resolution of SVHN, i.e., $(32, 32)$, then we use features extracted from the last layer of a LeNet5. We refer readers to Seguy et al. (2017) and Struckmeier et al. (2023) for further details on these benchmarks.

Our main point of comparison is with other OT-based DA methods. We compare our GMM-OTDA strategies with other OT methods for DA, namely, we consider the OT for DA (OTDA) strategy of Courty et al. (2017) (Exact and Sinkhorn), the linear mapping estimation of Flamary et al. (2019), and the InfoOT strategy of Chuang et al. (2023) (barycentric and conditional mappings). For the scalability experiment using the *MNIST, USPS, SVHN* benchmark, we consider the large scale OT methods of Seguy et al. (2017), denoted as Alg. 1 and 2. Furthermore, for completeness, we consider the Linearly Alignable Optimal Transport (LaOT) strategy of Struckmeier et al. (2023).

### 5.1 Scalability with respect $d$

In this section, our goal is to evaluate how our method scales with the data dimensionality $d$. To do so, we evaluate methods based on their performance on visual adaptation benchmarks. The goal is to classify images into categories, based on $2048-$dimensional vectors from ResNets (He et al., 2016) fine-tuned on the source domain of each adaptation task. We summarize our results in Figure 4. The detailed results may be found in the appendix, i.e., Table 4.

Over Caltech-Office and ImageCLEF, our methods outperform other state-of-the-art methods. For Office 31 and Office-Home, Info-OT$_c$ and OTDA$_{affine}$ proved to be more effect than our methods, but ours still ranks second. For Office 31, the density estimation strategy of Info-OT$_c$ proves effective in finding a better map between the domains. For Office-Home, the baseline is already one of the best performing methods. An affine transformation is therefore sufficient for an effective adaptation.

Nonetheless, our method surpasses empirical OT over all tested benchmarks, especially preventing negative transfer in the Office-Home benchmark. Likewise, approximating class conditional distributions $P(X|Y)$ through GMMs proves effective over empirical OT. Indeed, our method improves over HOT-DA of El Hamri et al. (2022), which is based on empirical distributions.

These experiments prove that our method can effectively perform UDA between high-dimensional distributions. Note that, for these benchmarks, we use $2048-$dimensional vectors, which is by far the largest dimensionality values considered in this study.

### 5.2 Scalability with resepct $n$

In this section, we use the *MNIST, USPS, SVHN* benchmark. Our goal is to evaluate how our method scales with the number of samples $n$. As we discussed in Remark 3.1, a major advantage of the GMM formulation is reducing OT complexity from $\mathcal{O}(n^3 \log n)$ to $\mathcal{O}(K^3 \log K)$, i.e., we replace number of samples

---

[2]https://engineering.case.edu/bearingdatacenter

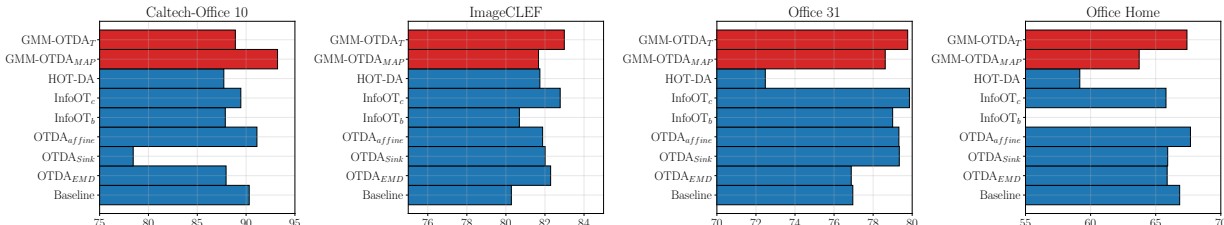

Figure 4: Average adaptation performance over 4 visual domain adaptation benchmarks.

by number of components, which are orders of magnitude inferior. To give a practical comparison, while MNIST has $n = 6 \times 10^5$ samples, we represent its probability distribution through a GMM with $K = 10^2$ components. This modeling choice improves scalability, especially when the components of GMMs have diagonal covariance matrices.

With respect performance, table 1 shows that GMM-OTDA has a similar performance to LaOT, i.e., performance degrades on $M \to U$, but increases on $U \to M$ and $M \to S$. Curiously, this corresponds to the case where a simpler dataset (e.g., MNIST) is transferred to a more complex dataset (e.g., SVHN). The performance similarity is not surprising, since LaOT and GMM-OTDA work under similar principles. However, the GMM modeling, again, proves superior to the Gaussian hypothesis, as we are able to improve performance on $U \to M$ and $M \to S$ tasks.

| Algorithm | $M \to U$ | $U \to M$ | $M \to S$ |
|---|---|---|---|
| Baseline | 73.47 | 36.97 | 54.33 |
| OTDA$_{EMD}$ | 57.75 | 52.46 | - |
| OTDA$_{Sink}$ | 68.75 | 57.35 | - |
| Alg. 1 of Seguy et al. (2017) with $H$ | 68.84 | 57.55 | 58.87 |
| Alg. 1 of Seguy et al. (2017) with $\ell_2$. | 67.80 | 57.47 | 60.56 |
| Alg. 1 + 2 of Seguy et al. (2017) with $H$ | **77.92** | 60.02 | 61.11 |
| Alg. 1 + 2 of Seguy et al. (2017) with $\ell_2$. | 72.61 | 60.50 | 62.88 |
| LaOT | 72.57 | 62.28 | 60.36 |
| GMM-OTDA (ours) | 71.83 | **63.11** | **87.19** |

Table 1: Large scale OT experiment. We consider the adaptation between 3 digit recognition benchmarks, namely, USPS, MNIST and SVHN. Overall, GMM-OTDA largely outperforms other methods on harder adaptation tasks, i.e., $U \to M$ and $M \to S$.

## 5.3 VisDA-C Benchmark

We experiment with the VisDA benchmark (Peng et al., 2017), a large scale DA dataset containing 152397 and 55388 source and target domain samples. As in the previous benchmarks, we pre-train the feature extractor using source domain data, then proceed to perform adaptation over the extracted features. This experiment stresses the scalability of our strategy in comparison with empirical OT methods, since solving an OT problem over this benchmark would lead to a linear program with $n_S \times n_T = 8.44 \times 10^9$ variables. More dramatically, for running the barycentric map over this benchmark one would need to store $\gamma$, leading to $n_S \times n_T$ floating point coefficients, that is, approximately 270.11 GB of memory.

To cope with the sheer volume of data, we run empirical OT methods on a sub-sample of $n_S = n_T = 15000$ samples. Parametric versions of OT methods, such as OTDA$_{affine}$ and GMM-OTDA are run with the full datasets, which illustrates the advantage of having a compact representation for distributions. Besides, we explore how these methods improve performance over different feature extractors. Hence, besides ResNet 50 and 101, we also consider a ViT-16-b (Dosovitskiy et al., 2021). Our results are shown in table 2.

| Algorithm | ResNet 50 | ResNet101 | ViT-b-16 |
|---|---|---|---|
| Source-Only | 47.93 | 53.90 | 56.70 |
| $\text{OTDA}_{\text{EMD}}$ | 53.69 $(\Delta + 5.76)$ | 57.42 $(\Delta + 3.52)$ | 63.25 $(\Delta + 6.55)$ |
| $\text{OTDA}_{\text{Sinkhorn}}$ | 53.02 $(\Delta + 5.09)$ | 10.54 $(\Delta - 43.36)$ | 66.75 $(\Delta + 10.05)$ |
| $\text{OTDA}_{\text{Affine}}$ | 7.46 $(\Delta - 40.47)$ | 11.82 $(\Delta - 42.08)$ | 6.75 $(\Delta - 49.95)$ |
| $\text{OTDA}_{\text{Affine-Diag}}$ | 51.41 $(\Delta + 3.48)$ | 56.91 $(\Delta + 3.01)$ | 59.94 $(\Delta + 3.24)$ |
| HOTDA | 47.51 $(\Delta - 0.42)$ | 47.64 $(\Delta - 6.26)$ | 62.55 $(\Delta + 5.85)$ |
| $\text{GMM-OTDA}_{T}$ | **58.35** $(\Delta + 10.42)$ | 56.57 $(\Delta + 2.67)$ | 74.10 $(\Delta + 17.40)$ |
| $\text{GMM-OTDA}_{\text{MAP}}$ | 57.36 $(\Delta + 9.43)$ | **58.77** $(\Delta + 4.87)$ | **74.44** $(\Delta + 17.74)$ |

Table 2: Comparison of domain adaptation performance over different feature extractors pre-trained with source domain data. We report the classification accuracy (in %) and the difference $\Delta$ over the source-only baseline. Our methods GMM-OTDA T and MAP consistently outperform other OT-based methods.

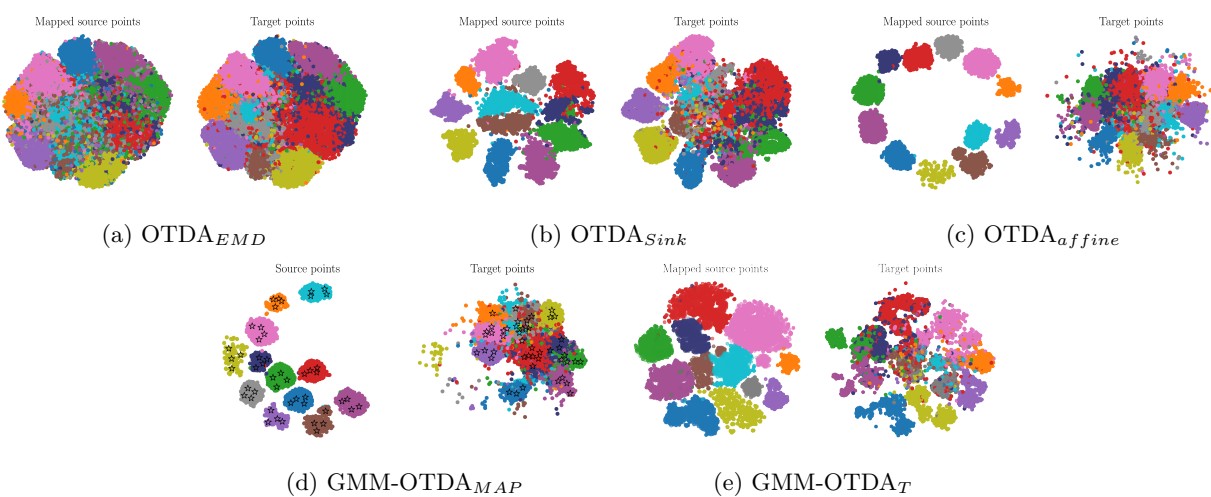

Figure 5: t-SNE visualization OT map-based DA techniques on the VisDA-C benchmark with ViT-16-b features. Colors represent different classes. In (d), stars represent the components of source and target GMMs, obtained through the conditional EM, and the EM algorithm respectively.

From table 2, note that our GMM-OTDA methods consistently outperform other OT-based methods. Furthermore, we ablate on the choice of using diagonal covariances, by comparing the performance of estimating an OT map between Gaussian measures with full (equation 8) and diagonal covariances (equation 10). Hence, using diagonal covariances provide a regularizing effect, improving the estimation of an OT map between Gaussian measures when the full-covariances are singular.

Finally, we analyze how the different mapping strategies match source and target domain measures. We summarize this analysis in Figure 5, where we show the t-Stochastic Neighbor Embeddings (SNE) of the concatenation of mapped source and target domain data. In contrast with $\text{OTDA}_{EMD}$, $\text{OTDA}_{Sinkhorn}$ and $\text{GMM-OTDA}_T$, $\text{OTDA}_{affine}$ does not manage to match source and target data, mainly due the simplicity of the Gaussian assumption. Furthermore, $\text{GMM-OTDA}_T$ manages to map source domain data in a way that does not mixes the classes (for instance, compare Figure 5 (a) with (e)). The label propagation approach is also discriminative of target domain classes, as is evidenced in Figure 5 (d). These considerations explain the superior performance of $\text{GMM-OTDA}_T$ with ViT-16-b features. We provide a similar analysis for the CWRU benchmark in the appendix.

## 6 Conclusion

In this paper, we consider the GMMOT framework of Delon & Desolneux (2020) as a candidate for UDA. Based on probability and OT theory, we devise 2 new effective strategies for UDA. The label propagation interprets the OT plan between GMM components as a joint probability distribution over source-target component pairs. This modeling choice allows us to predict the label of target GMM components through a label propagation equation similar to Redko et al. (2019). Furthermore, we propose a mapping strategy that transports samples from the same component together through an affine map, which has 2 advantages. First, it enforces group sparsity (Courty et al., 2017). Second, it has an analytical form in terms of GMM parameters. We show through a series of 85 UDA tasks that our methods outperform, or are competitive with the state-of-the-art in shallow domain adaptation, while being scalable with both number of samples $n$, and number of dimensions $d$. Our work further confirms previous studies on the intersection of GMMs and UDA, such as Montesuma et al. (2024b), showing that the GMMOT is a powerful candidate for shallow domain adaptation.

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

# A    Additional Details about Benchmarks

| Benchmark | Domains | Backbone | # Samples | # Classes |
|---|---|---|---|---|
| ImageCLEF | Caltech (C) | ResNet 50 | 600 | 12 |
| | Bing (B) | | 600 | |
| | ImageNet (I) | | 600 | |
| | Pascal (P) | | 600 | |
| | Total | | 2400 | |
| Caltech-Office 10 | Amazon (A) | ResNet 101 | 958 | 10 |
| | dSLR (D) | | 157 | |
| | Webcam (W) | | 295 | |
| | Caltech (C) | | 1123 | |
| | Total | | 2533 | |
| Office 31 | Amazon (A) | ResNet 50 | 2817 | 31 |
| | dSLR (D) | | 498 | |
| | Webcam (W) | | 795 | |
| | Total | | 4110 | |
| Office-Home | Art (Ar) | ResNet 101 | 2427 | 65 |
| | Clipart (Cl) | | 4365 | |
| | Product (Pr) | | 4439 | |
| | Real World (Rw) | | 4357 | |
| | Total | | 15588 | |

(a) Visual Domain Adaptation Benchmarks

| Benchmark | Domains | Backbone | # Samples | # Classes |
|---|---|---|---|---|
| CWRU | 1772rpm (A) | MLP | 8000 | 10 |
| | 1750rpm (B) | | 8000 | |
| | 1730rpm (C) | | 8000 | |
| | Total | | 24000 | |
| TEP | Mode 1 | Fully Convolutional | 2900 | 29 |
| | Mode 2 | | 2845 | |
| | Mode 3 | | 2899 | |
| | Mode 4 | | 2865 | |
| | Mode 5 | | 2883 | |
| | Mode 6 | | 2897 | |
| | Total | | 17289 | |
| CSTR | $N = 1.0, \epsilon = 0.00$ | - | 1300 | 13 |
| | $N = 1.0, \epsilon = 0.10$ | | 260 | |
| | $N = 1.0, \epsilon = 0.15$ | | 260 | |
| | $N = 0.5, \epsilon = 0.15$ | | 260 | |
| | $N = 1.5, \epsilon = 0.15$ | | 260 | |
| | $N = 2.0, \epsilon = 0.15$ | | 260 | |
| | Total | | 2860 | |

(b) Cross-Domain Fault Diagnosis Benchmarks

Table 3: Overview of Visual Domain Adaptation and Cross-Domain Fault Diagnosis benchmarks

In table 3, we show an overview of the used benchmarks. We run our experiments on 4 visual DA datasets, and 3 CDFD datasets. For vision, we use Residual Networkss (ResNets) pre-trained on ImageNET as the backbone. For each adaptation task (e.g., $C \rightarrow B$ in ImageCLEF) we fine tune the network using labeled source domain data. For all methods, we extract the features of source, and target domain, using the fine-tuned checkpoint. The size of the ResNet is used to agree with previous research, such as Peng et al. (2019) and Montesuma et al. (2023). We refer readers to[3] for further technical details on the fine-tuning of vision backbones.

For CDFD, we considered the same setting as previous works using these benchmarks, such as Montesuma et al. (2023), Montesuma et al. (2024c) and Montesuma et al. (2022), for Case Western Reserve University (CWRU), Tennessee Eastman (TE) process and Continuous Stirred Tank Reactor (CSTR) respectively. For CWRU, we extract windows out of raw signals of size 2048, then get the frequency representation for these windows using a fast Fourier transform. These are treated as 2048 feature vectors, that are then fed to a neural network. In the TE process, we consider the same setting of Montesuma et al. (2024c), i.e., we use a fully convolutional neural net. For the CSTR, we directly use the signals, concatenated into a $1400-$dimensional vector as the features. The data for these benchmarks is publicly available here[4], and here[5].

Note that, for each dataset, there are $n_{domains}(n_{domains} - 1)$ adaptation tasks, except for CSTR, which is a multi-target benchmark (i.e., a single source, and 6 targets). As a result, we have $12 \times 3 + 6 \times 2 + 30 + 6 = 84$ adaptation tasks.

---

[3] https://github.com/eddardd/DA-baselines
[4] https://www.kaggle.com/datasets/eddardd/tennessee-eastman-process-domain-adaptation
[5] https://www.kaggle.com/datasets/eddardd/continuous-stirred-tank-reactor-domain-adaptation

# B  Additional Experiments

## B.1  Detailed Results

| Benchmark | Task | Baseline | $\text{OTDA}_{EMD}$ | $\text{OTDA}_{Sink}$ | $\text{OTDA}_{affine}$ | $\text{InfoOT}_b$ | $\text{InfoOT}_c$ | HOT-DA | $\text{GMM-OTDA}_{MAP}$ | $\text{GMM-OTDA}_T$ |
|---|---|---|---|---|---|---|---|---|---|---|
| | $A \to D$ | 87.10 | 77.42 | 87.10 | 93.55 | 93.55 | 83.87 | **96.77** | 93.55 | 80.65 |
| | $A \to W$ | 91.53 | 93.22 | **96.61** | **96.61** | **96.61** | 94.92 | **96.61** | 93.22 | 91.53 |
| | $A \to C$ | 88.44 | **91.56** | 73.33 | 91.11 | 87.11 | 90.67 | 74.67 | 88.44 | 88.44 |
| | $D \to A$ | 88.54 | 90.10 | 93.23 | 92.19 | 91.15 | 92.71 | **96.88** | 96.35 | 95.83 |
| | $D \to W$ | 98.31 | 93.22 | 93.22 | 94.92 | **100.00** | 91.53 | 94.92 | **100.00** | 98.31 |
| | $D \to C$ | 75.56 | 67.11 | 17.33 | 71.56 | 68.44 | 75.11 | 51.11 | **86.67** | 71.56 |
| Caltech-Office | $W \to A$ | 85.94 | 82.81 | 15.62 | 81.25 | 60.94 | 86.46 | 69.79 | **87.50** | 84.90 |
| | $W \to D$ | **100.00** | 93.55 | 93.55 | 96.77 | 90.32 | 93.55 | 90.32 | 96.77 | 93.55 |
| | $W \to C$ | 84.89 | 87.56 | 87.56 | **88.44** | 87.56 | 87.56 | 84.89 | 87.56 | 88.00 |
| | $C \to A$ | **98.44** | 96.88 | **98.44** | **98.44** | **98.44** | 96.88 | **98.44** | **98.44** | **98.44** |
| | $C \to D$ | 93.55 | 87.10 | 90.32 | 93.55 | 87.10 | 87.10 | 100.0 | 93.55 | 80.65 |
| | $C \to W$ | 91.53 | 94.92 | 94.92 | 94.92 | 93.22 | 93.22 | 98.31 | **96.61** | 94.92 |
| | Avg. | 90.32 | 87.95 | 78.44 | 91.11 | 87.87 | 89.46 | 87.72 | **93.22** | 88.90 |
| | $B \to C$ | 92.50 | 95.00 | 95.00 | 94.17 | **97.50** | 96.67 | 96.67 | 95.00 | 94.17 |
| | $B \to I$ | 90.00 | 89.17 | 89.17 | 91.67 | 94.17 | 91.67 | **95.00** | **95.00** | 93.33 |
| | $B \to P$ | 68.33 | 69.17 | 70.00 | 71.67 | 73.33 | **75.83** | 74.17 | 72.50 | 74.17 |
| | $C \to B$ | 65.00 | 65.83 | 65.83 | 65.00 | 51.67 | 62.50 | 62.50 | 65.00 | **66.67** |
| | $C \to I$ | 89.17 | 96.67 | 96.67 | 95.00 | 92.50 | **97.50** | 95.83 | 94.17 | 96.67 |
| | $C \to P$ | 71.67 | 74.17 | 73.33 | 71.67 | 75.00 | **75.83** | 72.50 | 70.83 | **75.83** |
| ImageCLEF | $I \to B$ | 68.33 | **70.00** | 68.33 | **70.00** | 65.00 | 66.67 | 61.67 | 67.50 | **70.00** |
| | $I \to C$ | 93.33 | 95.83 | 95.83 | 95.83 | 95.83 | **96.67** | 95.83 | 95.00 | 95.83 |
| | $I \to P$ | 71.67 | 74.17 | **75.00** | 73.33 | 73.33 | 71.67 | 72.50 | 73.33 | **75.00** |
| | $P \to B$ | 67.50 | **69.17** | 66.67 | 68.33 | 57.50 | 65.83 | 62.50 | 62.50 | 64.17 |
| | $P \to C$ | 95.00 | 95.83 | 95.83 | 95.00 | **96.67** | **96.67** | **96.67** | 95.00 | 95.00 |
| | $P \to I$ | 90.83 | 92.50 | 92.50 | 90.83 | **95.83** | **95.83** | 95.00 | 94.17 | 95.00 |
| | Avg. | 80.28 | 82.29 | 82.01 | 81.88 | 80.69 | 82.78 | 81.74 | 81.67 | **82.99** |
| | $A \to D$ | 66.07 | 68.75 | 69.64 | 69.64 | 75.89 | **76.79** | 72.32 | 69.64 | 72.32 |
| | $A \to W$ | 76.02 | 74.27 | 80.12 | 80.12 | 79.53 | 79.53 | 73.68 | 76.61 | **80.70** |
| | $D \to A$ | 65.68 | 65.85 | 67.77 | 66.90 | 67.60 | 66.20 | 61.15 | 68.29 | **73.52** |
| Office 31 | $D \to W$ | 94.15 | 95.32 | 98.25 | 98.25 | 95.91 | 97.08 | 84.80 | **98.83** | 95.32 |
| | $W \to A$ | 63.41 | 66.90 | 67.42 | 65.51 | **67.60** | **67.60** | 61.67 | 66.38 | 65.68 |
| | $W \to D$ | **96.43** | 90.18 | 92.86 | 95.54 | 87.50 | 91.96 | 81.25 | 91.96 | 91.07 |
| | Avg. | 76.96 | 76.88 | 79.34 | 79.32 | 79.00 | **79.86** | 72.48 | 78.62 | 79.77 |
| | $Ar \to Cl$ | 55.10 | 54.98 | 54.87 | 56.24 | 17.41 | 53.95 | 47.88 | 53.95 | 57.96 |
| | $Ar \to Pr$ | 70.95 | 68.69 | 71.96 | 71.96 | 30.97 | 70.27 | 67.23 | 74.89 | 74.10 |
| | $Ar \to Rw$ | 79.68 | 79.68 | 80.83 | 80.71 | 40.53 | 80.25 | 76.00 | 77.96 | 82.43 |
| | $Cl \to Ar$ | 63.51 | 60.62 | 63.09 | 62.68 | 31.34 | 62.68 | 53.81 | 59.79 | 64.33 |
| | $Cl \to Pr$ | 69.26 | 66.89 | 68.81 | 70.72 | 41.78 | 68.92 | 63.51 | 70.05 | 71.73 |
| | $Cl \to Rw$ | 72.68 | 69.92 | 71.18 | 72.33 | 38.81 | 71.18 | 67.97 | 68.66 | 74.63 |
| Office-Home | $Pr \to Ar$ | 66.80 | 62.47 | 64.12 | 66.39 | 32.16 | 64.33 | 55.88 | 57.73 | 62.89 |
| | $Pr \to Cl$ | 36.88 | 38.83 | 25.32 | 38.83 | 8.59 | 30.93 | 23.71 | 30.70 | 31.62 |
| | $Pr \to Rw$ | 78.76 | 77.84 | 79.22 | 79.22 | 47.99 | 78.30 | 71.64 | 73.59 | 80.94 |
| | $Rw \to Ar$ | 72.99 | 71.96 | 72.37 | 73.81 | 51.13 | 70.72 | 62.47 | 66.39 | 69.69 |
| | $Rw \to Cl$ | 53.15 | 57.85 | 57.39 | 56.93 | 37.00 | 56.59 | 47.65 | 50.63 | 56.36 |
| | $Rw \to Pr$ | 82.32 | 80.86 | 81.87 | 82.21 | 64.86 | 81.31 | 72.30 | 80.41 | 82.09 |
| | Avg. | 66.84 | 65.88 | 65.92 | 67.67 | 36.88 | 65.79 | 59.17 | 63.73 | 67.40 |

Table 4: Single-source domain adaptation results. We compare 8 methods over 5 benchmarks, with a total of 42 adaptation tasks.

## B.2  Ablations and Visualization

**Ablating the number of components and entropic regularization.** In this experiment, we ablate the two parameters of our methods, namely, the number of components $K$, and the entropic regularization $\epsilon$. Recall that we normalize the ground-cost by the maximum value, i.e., $\tilde{C}_{ij} = C_{ij}/(\max_{ij} C_{ij})$, which improves the numerical stability of the Sinkhorn algorithm. We summarize our results in Figure 6.

For the number of components $K$, the relationship with performance is mostly clear. Indeed, except for $\epsilon = 10^{-1}$ on the mapping strategy, *using more components enhances performance*. Note that, even though this implies a more complex GMMs, we still have far less components than samples ($n = 600$ per domain, i.e., 5 times more samples than components).

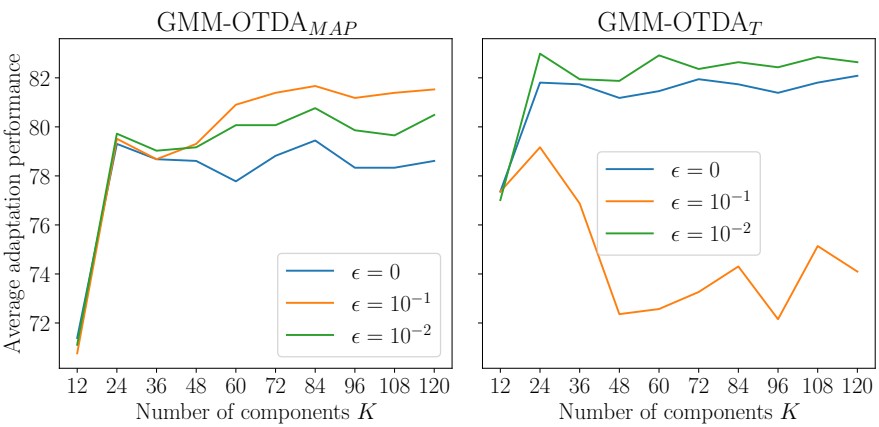

Figure 6: Ablation on number of components $K$, and entropic penalty $\epsilon$, for the MAP estimation strategy based on labeled propagation (left), and the mapping estimation strategy (right).

For the entropic penalty $\epsilon$, we have two drastically different scenarios. For the MAP estimation, using higher entropic regularization coefficients improve performance, whereas the mapping strategy works better for smaller regularization coefficients (or exact OT). While this may seem contradictory, we note that, in the mapping strategy, we actually filter out irrelevant matchings between components based on a parameter $\tau$. However, the entropic regularization is known to generate smoother couplings, which means that more entries of $\omega$ are non-zero, or possibly greater than a fixed $\tau$. As a consequence, the mapping strategy ends up behaving like $T_{rand}$, which causes a bad reconstruction for the target domain. Naturally, this effect gets amplified with more components in both GMMs, as there are more possible matchings.

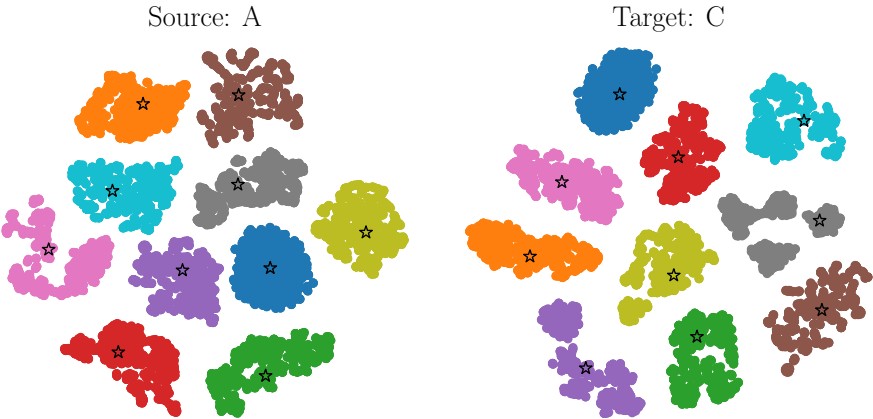

Figure 7: Source and target domain samples alongside the centroids (denoted by stars) found through EM. colors reflect the different classes.

**Visualizing components and mapped samples.** In this experiment, we use the CWRU benchmark adaptation task $A \rightarrow C$. We start by embedding the source and target domain data with the t-SNE technique of Van der Maaten & Hinton (2008). We do this in 2 separate plots, where we concatenate the source domain features with the centroids obtained by running the EM algorithm (resp. target). This visualization is shown in Figure 7. Next, we map samples from the source to the target domain, with the various strategies described in this section, with the exception of InfoOT, which did not had a reasonable running time. These are shown in Figure 8.

Overall, while the exact OT solution provides a measure that better reflects the feature positions, it mixes the classes, as evidenced by the orange and blue classes being mapped to the same place, as well as the green and violet classes. This phenomenon does not happen for other methods, at the cost of having mapped

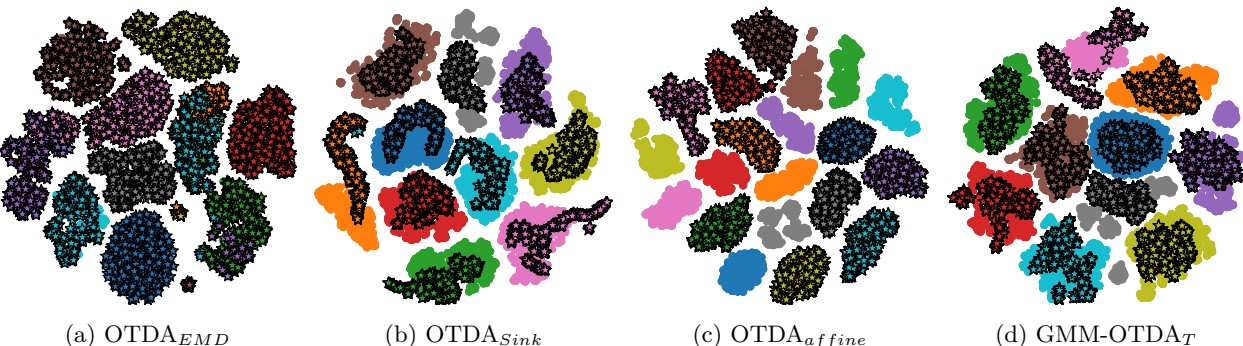

| (a) OTDA$_{EMD}$ | (b) OTDA$_{Sink}$ | (c) OTDA$_{affine}$ | (d) GMM-OTDA$_T$ |

Figure 8: t-SNE visualisation of mapped samples to the target domain, on the CWRU task $A \rightarrow C$.

points distributed in a different way. However, for OTDA$_{sink}$ and GMM-OTDA$_T$, the mapped points better respect the class boundaries. For OTDA$_{affine}$, note that the mapped distribution does not actually match the target. Overall, we achieve a better mapping through the GMM modeling.

### B.3   Cross-Domain Fault Diagnosis

For CDFD benchmarks, we follow the experimental settings of Montesuma et al. (2023), Montesuma et al. (2022) and Montesuma et al. (2024c), which roughly follows a similar idea to visual adaptation tasks. This means that we pre-train a neural network with source domain data, then use its encoder for feature extraction. We refer readers to the original papers, and our appendix, for further information. To summarize our experimentation, there are in total 8 benchmarks, and 84 domain adaptation tasks.

**Case Western Reserve University Benchmark.** With respect other benchmarks, the CWRU has the most number of samples per domain, i.e., 8000. In this case, InfoOT was intractable due its computational complexity. Furthermore, this benchmark illustrate the advantage of employing a grouping technique for enforcing the class structure in OT.

| Task | Baseline | OTDA$_{EMD}$ | OTDA$_{Sink}$ | OTDA$_{affine}$ | InfoOT$_b$ | InfoOT$_c$ | HOT-DA | GMM-OTDA$_{MAP}$ | GMM-OTDA$_T$ |
|---|---|---|---|---|---|---|---|---|---|
| $A \rightarrow B$ | 51.12 | 72.00 | 75.19 | 78.12 | - | - | 69.88 | **79.75** | **79.75** |
| $A \rightarrow C$ | 62.88 | 94.12 | **100.00** | 95.62 | - | - | **100.00** | 99.94 | **100.00** |
| $B \rightarrow A$ | 42.50 | 76.12 | 78.50 | 75.88 | - | - | 79.75 | **80.00** | **80.00** |
| $B \rightarrow C$ | 37.44 | 77.62 | 78.88 | 75.38 | - | - | 79.81 | 79.56 | **79.94** |
| $C \rightarrow A$ | 52.81 | 98.38 | 99.25 | 94.12 | - | - | 98.75 | 99.12 | **99.88** |
| $C \rightarrow B$ | 55.62 | 70.25 | 74.50 | 75.50 | - | - | 83.12 | 79.75 | **80.00** |
| Avg. | 50.40 | 81.42 | 84.39 | 82.44 | - | - | 85.22 | 86.35 | **86.59** |

Table 5: Experimental results on the CWRU benchmark. For each task (i.e., each row), we highlight the best performing method in bold. Overall, InfoOT did not have a reasonable running time due the large number of samples on each domain.

**Continuous Stirred Tank Reactor.** As covered in Montesuma et al. (2022), the CSTR process carries an exothermic reaction $A \rightarrow B$. The reactor is jacketed, and an inflow of coolant is controlled by a Proportional, Integral, Derivative (PID) controller as described in Pilario & Cao (2017). From this reactor, a set of 7 variables are measured throughout simulation, corresponding to different temperatures, concentrations and flow-rates. We refer readers to Montesuma et al. (2022) for further details. Associated with this process, there are a set of 12 different faults, ranging from process and sensors faults, and input disturbances. On top of these 12 faults, there is the no-fault scenario, characterizing a classification problem with 13 classes.

The different domains in this benchmark correspond to changes in the process conditions. These are of 2 kinds. First, one introduces a noise, $\eta$, in the process parameters (e.g., reactor or jacket volume), reflecting the possible uncertainty in the mathematical modeling of the reactor. Second, one changes the reaction

| Target Domain | 1 | 2 | 3 | 4 | 5 | 6 | Score |
|---|---|---|---|---|---|---|---|
| Reaction Order ($N$) | 1.0 | 1.0 | 1.0 | 0.5 | 1.5 | 2.0 | |
| Parameter Noise ($\eta$) | 10% | 15% | 20% | 15% | 15% | 15% | |
| Baseline | 69.23 | 67.30 | **73.07** | 53.84 | 63.46 | **57.69** | 64.10 |
| OTDA$_{EMD}$ | 71.15 | 71.15 | 71.15 | 61.53 | 57.69 | 50.00 | 63.78 |
| OTDA$_{Sink}$ | 67.31 | 67.31 | 69.23 | 55.76 | 51.92 | 53.84 | 60.89 |
| OTDA$_{Affine}$ | 65.38 | 71.15 | 71.15 | 61.54 | **65.38** | 53.84 | 64.74 |
| InfoOT$_b$ | 67.31 | 67.31 | 67.31 | 50.00 | 51.92 | 40.07 | 58.65 |
| InfoOT$_c$ | 71.15 | 67.31 | 71.15 | 53.84 | 51.92 | 48.07 | 60.57 |
| HOT-DA | 55.77 | 40.38 | 44.23 | 55.77 | 40.38 | 42.31 | 46.47 |
| GMM-OTDA$_{MAP}$ | **78.84** | **73.07** | **73.07** | 61.54 | 57.69 | 55.77 | **66.67** |
| GMM-OTDA$_T$ | 76.92 | **73.07** | **73.07** | **63.46** | 53.84 | 50.00 | 65.04 |

Table 6: Average classification accuracy with confidence intervals over a 5-fold cross-validation.

order, $N$, of the reaction $A \to B$, which drastically changes the dynamics of the system. As Montesuma et al. (2022), we consider $\eta \in \{0.1, 0.15, 0.2\}$, and $N \in \{1, 0.5, 1.5, 2\}$. The source domain corresponds to $N = 1$, $\eta = 0.0$, whereas the 6 different targets correspond to combinations of $\eta$ and $N$. These are shown in Table 6.

The CSTR benchmark poses a few challenges. First, it has a small number of samples on domains. While the source is composed of 1300 samples, each target only has 260. Second, each sample lies in a $1400-$dimensional space. Third, target domains are noisy, due the parameter noise $\eta$. As a result, most methods have difficulty in adapting, and performance usually degrades for more intense shifts (e.g., $N = 2.0$ and $\eta = 0.15$). However, GMM-OTDA$_{MAP}$ and GMM-OTDA$_T$ outperform other methods.

**Tennessee Eastman Process.** Our last experiment consists of the TE process, a benchmark widely used by the chemical engineering community (Reinartz et al., 2021). This benchmark has the largest number of domains, i.e., 6. Each of these domains is characterized by a different mode of production for the products of a chemical reaction, which affects the measured signals from the chemical plant. We summarize our results in Figure 9, which comprises the 30 adaptation tasks.

For this benchmark, we compare 11 methods. We divide those into *shallow* DA methods, and *deep* DA methods. Shallow methods try to cope with distributional shift by transforming or re-weighting the samples in a feature space. In the case of this benchmark, we use the encoder's activations as features. In contrast, deep methods cope with distribution shift by learning discriminative, *domain invariant* features, by penalizing the encoder's parameters $\theta_g$ so that, after encoding the data points, the domains are indistinguishable from each other. Besides the 6 shallow methods compared throughout this paper, we also consider classic deep methods, such as Domain Adversarial Neural Network (DANN) (Ganin et al., 2016), Domain Adaptive Network (DAN) (Ghifary et al., 2014), Wasserstein Distance Guided Representation Learning (WDGRL) (Shen et al., 2018) and Deep Joint Distribution Optimal Transport (DeepJDOT) (Damodaran et al., 2018).

In comparison with other benchmarks, the TE feature vectors have fewer dimensions (i.e., 128). As a result, OTDA$_{EMD}$ is the best performing method. However, GMM-OTDA manages to improve over OTDA$_{affine}$ and HOT-DA. Overall, our methods are especially better on harder adaptation tasks, such as $6 \to 2$ and $3 \to 2$. We refer readers to the exploratory data analysis of Montesuma et al. (2024c) for further insights on why these adaptation tasks are harder.

Furthermore, note that shallow methods are comparatively better to deep methods. Indeed, the deep neural nets use considerably less data than, for instance, the image benchmarks considered in our experiments section. In this latter case, previously to the fine-tuning step on the source domain data, ResNets and ViTs are pre-trained on the ImageNet benchmark (Deng et al., 2009), which provides a good starting model for natural image classification. In the context of the TE process benchmark, this is not possible since data are time series of a specific chemical process.

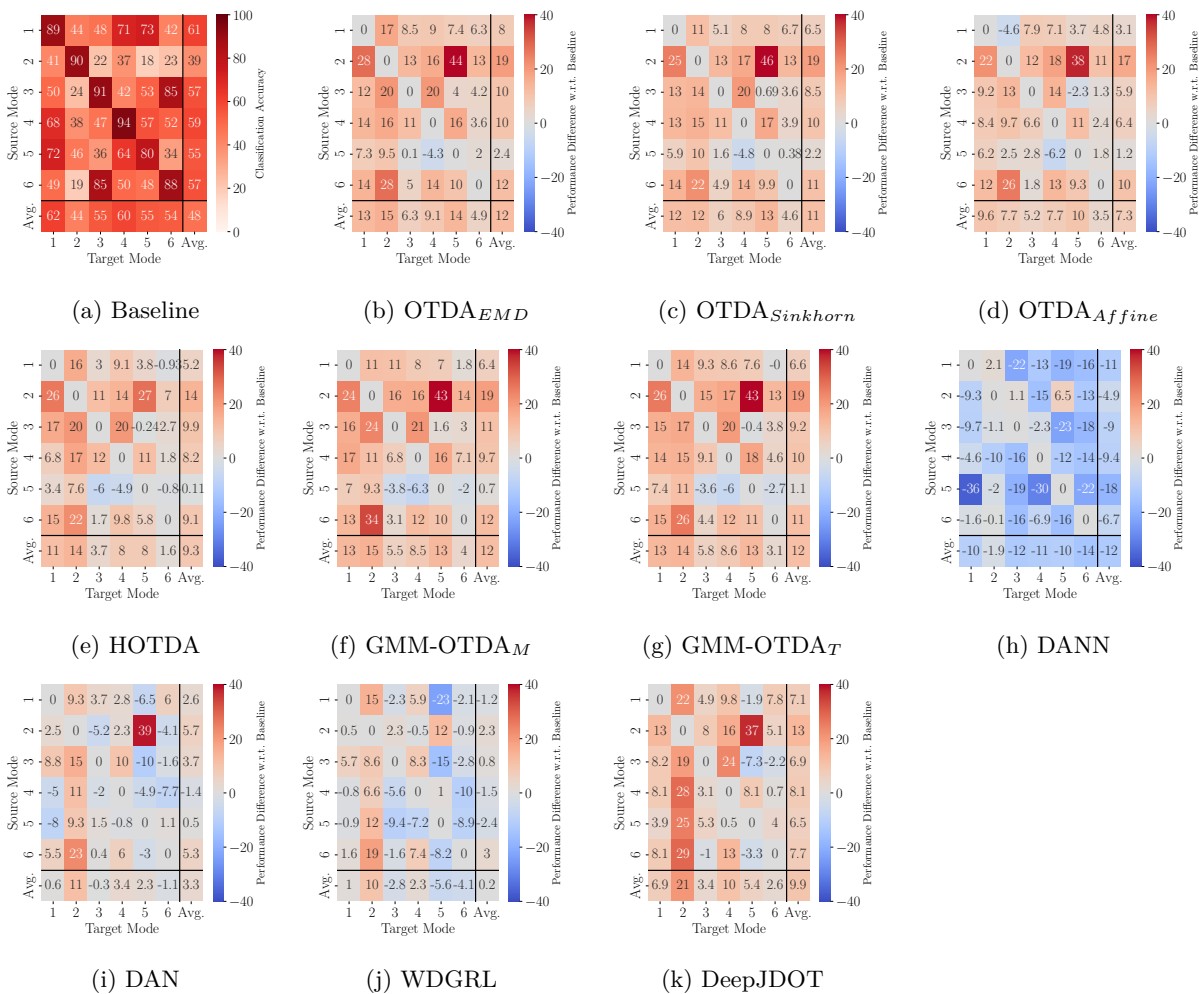

Figure 9: Domain adaptation results on the Tennessee Eastman Process. In (a), we show the baseline adaptation tasks, where each row represents a source domain, and each column represents a target domain. From (b) to (k), we show the performance offset with respect (a) of adaptation algorithms. Note that (i) through (k) are deep learning-based algorithms.

## B.4 Running Time Analysis

Besides our remark 3.1, we also run a running time analysis of the tested algorithms. Our experimental setting is as follows. We use the adaptation task $A \to W$ of the Office 31 benchmark. In this case, $n_S = 2817$, $n_T = 624$, $n_c = 31$ and $d = 2048$. We ran each algorithm 10 independent times, then computed the mean and standard deviation of their running time. Our results are reported on tables 7 and 8.

| Algorithm | Running Time (seconds) | Accuracy (%) |
|---|---|---|
| $OTDA_{EMD}$ | $0.775 \pm 0.007$ | 74.27 |
| $OTDA_{Sink}$ | $14.119 \pm 0.125$ | 80.12 |
| $OTDA_{Affine}$ | $8.503 \pm 0.097$ | 80.12 |
| InfoOT | $105.407 \pm 0.716$ | 79.53 |
| HOTDA | $2.508 \pm 0.023$ | 73.68 |

Table 7: Running time (in seconds) and classification accuracy (in %) of different OT-based strategies.

Starting from table 7, the fastest algorithm is $\text{OTDA}_{\text{EMD}}$, which has complexity $\mathcal{O}(n^3 \log n)$. In comparison, $\text{OTDA}_{\text{Sink}}$ has complexity $\mathcal{O}(n^2)$ per iteration. Here, it is important to note that we run the Sinkhorn algorithm until convergence, for 1000 iterations, which explains its superior running time. It is noteworthy that $\text{OTDA}_{affine}$ also has a higher running time, since its complexity is dimension-dependent, i.e., $\mathcal{O}(d^3)$. Due the high dimensional character of the data at hand, this results in a higher running time.

Another example of higher running time comes from HOTDA, which solves $n_c^2 - n_c = 465$ sub empirical OT problems, resulting in a higher running time in comparison with $\text{OTDA}_{\text{EMD}}$. Finally, it is noteworthy that InfoOT is considerably slower than other methods, due to its $\mathcal{O}(n^3)$ complexity *by iteration*.

In comparison with previous methods, we show the running time and classification accuracy of $\text{GMM-OTDA}_{\text{T}}$ and $\text{GMM-OTDA}_{\text{MAP}}$ for $K = \{31, 62, \cdots, 217\}$. We do so for $\epsilon = 10^{-2}$, which in practice yielded the best empirical performance. As a result, the running time should be directly compared to $\text{OTDA}_{\text{Sink}}$. For all number of components, our algorithm has an inferior running time to almost all methods, with the expection of $\text{OTDA}_{\text{EMD}}$ and HOTDA.

| Number of Components | Running Time (seconds) | GMM-OTDA$_{\text{T}}$ | GMM-OTDA$_{\text{MAP}}$ |
|---|---|---|---|
| 31 | $0.742 \pm 0.074$ | 78.94 | 64.91 |
| 62 | $1.329 \pm 0.015$ | 79.53 | 70.76 |
| 93 | $1.995 \pm 0.035$ | 76.02 | 70.17 |
| 124 | $2.679 \pm 0.051$ | 80.70 | 74.85 |
| 155 | $3.517 \pm 0.025$ | 78.36 | 75.44 |
| 186 | $4.343 \pm 0.090$ | 80.70 | 70.17 |
| 217 | $5.069 \pm 0.059$ | 77.77 | 76.02 |

Table 8: Running time (in seconds) and classification accuracy (in %) of GMM-OTDA$_{\text{T}}$ and GMM-OTDA$_{\text{MAP}}$ as a function of number of components.

