# OpenReview forum: "Optimal Transport for Domain Adaptation through Gaussian Mixture Models"
_TMLR — Accepted by TMLR_

### Review · Reviewer_zrmE · 2024-11-07

**Summary Of Contributions:**

This paper using optimal transport within Gaussian mixture models (GMMs) for solving the UDA problem. To achieve this, the propose method has two key aspects: 1. label propagation: propagate the ground truth knowledge of the source domain's GMM to that of the target domain using the optimal transport. 2. Sample Mapping: to ensure the effective transport of samples from the source domain to the target domain, a mapping estimation is proposed. The authors validate the proposed method via extensive experiments across 8 benchmarks. They demonstrate that their proposed method is competitive with existing state-of-the-art methods while maintaining efficiency in terms of both sample size and dimensionality.

**Audience:**

Yes

**Claims And Evidence:**

Yes

**Requested Changes:**

1. Providing a clear guidelines or instructions for practitioners on how to implement the proposed method, especially for those who might not be familiar with GMMs and OT.

2. Including a more detailed discussion on the limitations of OT and GMMs: (1) The assumptions made about data distributions; (2) The issues that can arise from misestimating the number of GMM components. (3) cases where the methods might not perform well, such as in high-dimensional spaces, with sparse data, or with distributions that do not fit well with GMMs.

3. Conducting a thorough ablation study to analyze how different hyperparameters (e.g., the number of components and covariance structure) affect performance.

4. Conducting experiments on larger datasets, such as VisDA-2017-C, to see how well the model performs and generalizes. This will provide insights into its scalability and robustness against overfitting.

5. Discussing the importance of the quality and representativeness of the source data. Suggesting ways to evaluate the quality of the source data and its potential impact on adapting to the target domain.

6. Conducting a runtime complexity analysis of the proposed method and compare it with other UDA methods.

**Strengths And Weaknesses:**

### Strengths

1. The proposed work is well-grounded in OT theory and GMMs, which provides a solid theoretical basis for the proposed method.

2. Using label propagation to explain the OT between the source domain and the target domain is interesting.

3. The proposed work addresses the computational challenges associated with traditional OT and GMM methods, particularly in high-dimensional spaces, by demonstrating a reduction in complexity when applying GMMs.

4. The paper is well-written with clear motivations. The experiments are also well-structured and clearly presented.

### Weaknesses:

1. While the theoretical contributions are strong, implementing the proposed method might be complicated, particularly with larger datasets. This complexity might make it challenging for practitioners who are not familiar with GMMs and OT.

2. Since OT requires working in spaces of the same dimension and GMMs have their own limitations, the paper could benefit from discussing these drawbacks in more detail. To be specific, it should discuss more about the assumptions made about data distributions, like the challenges that can arise if the number of components is misestimated. Moreover, the paper should consider scenarios where the methods might not perform well, such as with high dimensionality, sparse data, or distributions that do not fit well with GMMs.

3. The performance of GMMs can be sensitive to hyperparameters like the number of components and the covariance structure. The authors should conduct a more detailed ablation study to assist practitioners in selecting the appropriate hyperparameters.

4. GMMs can be prone to overfitting, especially if too many components are used relative to the amount of data available, which could lead to poor generalization on unseen data. Could the authors conduct the experiments on a larger dataset, like VisDA-2017-C?

5. The effectiveness of the proposed methods heavily relies on the quality and representativeness of the source data. If the source data is biased or unrepresentative, the adaptation may fail to generalize well to the target domain.

6. While the paper emphasizes efficiency, implementing GMMs and OT still demands considerable computational resources. Could the authors conduct a runtime complexity analysis and compare it with other UDA methods?

---

> ### Author Response · Authors · 2024-11-16
> **Rebuttal (1 / 2)**
>
> Dear Reviewer,
>
> Thank you for your feedback. We are pleased that you found our work to be __well grounded on a solid basis__, mainly OT and GMMs, as well as __interesting__. We are encouraged that you recognized that our work __addresses the computational challenges associated with traditional OT in high dimensions__, and that our paper is __well written__.
>
> In the following, we address your requested changes.
>
> > While the theoretical contributions are strong, implementing the proposed method might be complicated, particularly with larger datasets. This complexity might make it challenging for practitioners who are not familiar with GMMs and OT.
>
> Thank you for pointing that out. We will include in our revised manuscript a complete pseudo-code for our fitting strategy for the GMM on the source domain, as well an overall pseudo-code on the whole algorithm. We will also include some background in the appendix so that readers not familiar with these topics can familiarize themselves.
>
> > Including a more detailed discussion on the limitations of OT and GMMs: (1) The assumptions made about data distributions; (2) The issues that can arise from misestimating the number of GMM components. (3) cases where the methods might not perform well, such as in high-dimensional spaces, with sparse data, or with distributions that do not fit well with GMMs.
>
> 1. We assume that data distributions are multi-modal, as is commonly in classification. We further assume that these distributions can be represented through GMMs.
>
> 2. In principle, the number of components in the GMMs is a parameter of complexity. As we show in our Appendix B.2, using too few components usually leads to performance degradation due to the model not being complex enough. In general, we found that our methods are robust to the __over-estimation__ of the number of components (see B.2).
>
> 3. All of these cases pose challenges to our method, especially if we violate assumption (1) above.
>
> > Conducting a thorough ablation study to analyze how different hyperparameters (e.g., the number of components and covariance structure) affect performance.
>
> We refer the reviewer to our appendix B.2, where we ablate the number of components and the entropic regularization parameter. We further note that, in our experiments, we tried estimating full covariances, but in most datasets this is not feasible due to singular covariance matrices. We did a similar experiment on VisDA below. Here, it is important to note the advantage of a simpler covariance structure (i.e., diagonal covariances), as estimation is easier and computational complexity is reduced.
>
> > Discussing the importance of the quality and representativeness of the source data. Suggesting ways to evaluate the quality of the source data and its potential impact on adapting to the target domain.
>
> In domain adaptation, the quality and similarity of source domain data influences the final adaptation performance, such as discussed by (Ben-David et al., 2010). We will include further discussion about this point in the final version of our manuscript.
>
> # References
>
> Ben-David, S., Blitzer, J., Crammer, K., Kulesza, A., Pereira, F., & Vaughan, J. W. (2010). A theory of learning from different domains. Machine learning, 79, 151-175.

---

> ### Author Response · Authors · 2024-11-16
> **Rebuttal (2 / 2)**
>
> > Conducting experiments on larger datasets, such as VisDA-2017-C, to see how well the model performs and generalizes. This will provide insights into its scalability and robustness against overfitting.
>
> __Results on VisDA.__ We pre-train a ResNet-101 on the source domain data, then proceed to extract the source and target domain features for domain adaptation. Here are our results,
>
> | Algorithm | Accuracy on target domain |
> |-----------|---------------------------|
> |OTDA$\_{EMD}$|57.42|
> |OTDA$\_{Sinkhorn}$|10.54|
> |OTDA$\_{Affine}$|11.82|
> |HOTDA|47.64|
> |GMM-OTDA$\_{T}$|56.57|
> |GMM-OTDA$\_{MAP}$|58.77|
>
> It is important to clarify how OTDA and HOTDA were implemented on this benchmark. We subsample the source and target domain, selecting $n = 15000$ samples from each. This approach simplifies the adaptation problem, making it computationally feasible. The source domain data is then transported into the target domain, where it is used to train a classifier, and he trained classifier is subsequently used to classify all samples in the target domain. Performance is evaluated in terms of classification accuracy over the $n_{T} = 55388$ samples.
>
> Regarding the results in the table, OTDA$\_{sinkhorn}$ and OTDA$\_{Affine}$ perform significantly worse than the other methods. Specifically, the estimated covariances in the source and target domains are singular, making matrix inversion and square root calculations particularly challenging. Even by regularizing $\tilde{\Sigma}^{(P)} = \Sigma^{(P)} + \lambda I$ we could not get meaningful results.
>
> To demonstrate the effectiveness of axis-aligned Gaussian measures in such cases, we evaluated the affine map estimation using covariances of the form  $\Sigma = \text{diag}(\sigma\_{1},\cdots,\sigma\_{d})$, which leads to the map,
>
> $$T(\mathbf{x}) = \mathbf{Ax} + \mathbf{b}$$
>
> where $\mathbf{A} = \text{diag}(\sigma^{(Q)} / \sigma^{(P)})$ and $\mathbf{b} = \mu^{(Q)} - \mathbf{A}\mu^{(P)}$
>
> The axis-aligned covariance model achieved a significantly higher classification accuracy of $56.91\%$ compared to $11.82\%$ obtained using the affine map with full covariances.
>
> Overall, this experiment highlights two key observations:
>
> 1. Subsampling negatively impacts performance. Consequently, methods that efficiently represent probability measures, such as those using clustering, can better leverage a larger number of samples.
> 2. Axis-aligned covariance matrices provide a regularizing effect, improving the estimation of an OT map between Gaussian measures when the full-covariances are singular.
>
> __Running-Time Analysis.__ We performed a run-time analysis of our algorithm in comparison with the other tested algorithms with respect the adaptation task $A \rightarrow W$ on Office 31. In this case, $n\_{S} = 2817$, $n\_{T}=624$, $n\_{c} = 31$ and $d = 2048$. We ran each algorithm 10 independent times, then computed the mean and standard deviation of their running time.
>
>
> | Algorithm | Running Time (seconds) |
> |-----------|------------------------|
> |OTDA$\_{EMD}$|0.775 $\pm$ 0.007     |
> |OTDA$\_{Sink}$|14.119 $\pm$ 0.125   |
> |OTDA$\_{Affine}$|8.503 $\pm$ 0.097  |
> |InfoOT| 105.407 $\pm$ 0.716 	     |
> |HOTDA|2.508 $\pm$ 0.023 	     |
>
> Now, concerning our GMM-OTDA, the complexity depends on $K$. We report our running time with respect $K = 31, 62, 93, 124, 155, 186, 217$,
>
> | Number of components | Running Time (seconds) |
> |----------------------|------------------------|
> |31|   0.742 $\pm$ 0.074 |
> |62|   1.329 $\pm$ 0.015 |
> |93|   1.995 $\pm$ 0.035 |
> |124|  2.679 $\pm$ 0.051 |
> |155|  3.517 $\pm$ 0.025 |
> |186|  4.343 $\pm$ 0.090 |
> |217|  5.069 $\pm$ 0.059 |
>
> Overall, notice how our method is _faster_ than most of the previous methods, with the exception of OTDA$\_{EMD}$ and $HOTDA$. These results highlight our claim that our method improves the scalability of OT-based domain adaptation methods, mainly due to 2 factors,
>
> - The OT problem being solved is $K\_{P} \times K\_{Q}$ (number of components by number of components). Since $K\_{P}$ and $K\_{Q}$ are ideally much smaller than $n\_{S}$ and $n\_{T}$, we have a lower computational complexity while solving OT.
> - Furthermore, we alleviate the storage complexity of OTDA, which needs to store a $n\_{S} \times n\_{T}$ transport plan. For large datasets (such as MNIST, SVHN, or VisDA) this is prohibitive.

---

> ### Comment · Reviewer_zrmE · 2024-11-21
> **Official Comment by Reviewer zrmE**
>
> Thank you to the authors for the detailed response, which has addressed most of my concerns. However, as noted by Reviewer MxMx, TMLR recommends that authors submit a revised manuscript during the rebuttal period. I suggest that the authors submit a revised version of the manuscript that incorporates the changes we requested. Once I have reviewed the revised version, I can recommend the paper for acceptance.

---

> > ### Author Response · Authors · 2024-11-25
> > **Response to Official Comment by Reviewer zrmE**
> >
> > We are pleased that our rebuttal addressed your concerns. In the current phase, we included responses to Reviewers __MxMx__ and __hmXt__, including more experiments, as well as a revised version of our manuscript. Let us know if you have further questions.

---

> > > ### Comment · Reviewer_zrmE · 2024-11-26
> > > **Followup Comment by Reviewer zrmE**
> > >
> > > I would like to thank the authors for uploading the revised version. Can you highlight the revision in a different color other than black? Meanwhile, could you please specify on which page you have incorporated the changes I requested?

---

> > > > ### Author Response · Authors · 2024-11-27
> > > > **Response to followup comment by Reviewer zrmE**
> > > >
> > > > Dear Reviewer,
> > > >
> > > > We just uploaded an updated version with the changes highlighted in blue. Let us know if that is clearer than the previous version.
> > > >
> > > > Concerning your requested changes,
> > > >
> > > > > Providing a clear guidelines or instructions for practitioners on how to implement the proposed method, especially for those who might not be familiar with GMMs and OT.
> > > >
> > > > We added 4 algorithms, you may find them in pages 6 and 7. For further familiarization to GMMs, we added a reference to (M. Bishop, 2006; Chapter 9) in page 6.
> > > >
> > > > > Including a more detailed discussion on the limitations of OT and GMMs: (1) The assumptions made about data distributions; (2) The issues that can arise from misestimating the number of GMM components. (3) cases where the methods might not perform well, such as in high-dimensional spaces, with sparse data, or with distributions that do not fit well with GMMs.
> > > >
> > > > We added discussion on limitations in Page 7.
> > > >
> > > > > Conducting a thorough ablation study to analyze how different hyperparameters (e.g., the number of components and covariance structure) affect performance.
> > > >
> > > > The ablation is available in Appendix B.2, in Page 18.
> > > >
> > > > > Conducting experiments on larger datasets, such as VisDA-2017-C, to see how well the model performs and generalizes. This will provide insights into its scalability and robustness against overfitting.
> > > >
> > > > Results on VisDA are reported in section 5.3, Page 11
> > > >
> > > > > Discussing the importance of the quality and representativeness of the source data. Suggesting ways to evaluate the quality of the source data and its potential impact on adapting to the target domain.
> > > >
> > > > We added the discussion in Section 3.2, in between pages 5 and 6.
> > > >
> > > > > Conducting a runtime complexity analysis of the proposed method and compare it with other UDA methods.
> > > >
> > > > The runtime analysis is added in appendix B.4, on page 21.
> > > >
> > > > Please note that some of your requested changes are in the appendix, in order to keep the paper within 12 pages.

---

> > > > > ### Comment · Reviewer_zrmE · 2024-11-27
> > > > > **Decision on Recommendation**
> > > > >
> > > > > I would like to thank the authors for their effort during author response. All of my concerns have been satisfactorily addressed. Thus, I recommend accepting the paper for publication.

---

> > > > > > ### Author Response · Authors · 2024-11-30
> > > > > >
> > > > > > Dear reviewer,
> > > > > >
> > > > > > We are glad that our answers addressed your concerns. Thank you for your valuable feedback, which helped make our submission stronger.

---

### Review · Reviewer_MxMx · 2024-11-07

**Summary Of Contributions:**

This work proposes an optimal transport algorithm between Gaussian Mixture Models to address domain shift. The algorithm can efficiently scale with both the number of samples and dimensions.

**Audience:**

Yes

**Claims And Evidence:**

No

**Requested Changes:**

- The writing, especially regarding notations, requires an extensive revision. E.g.,
  - In Eq.(2) and Eq.(3), the right sides are the same while regarding the left sides, one is a joint distribution, and the other is a distance metric.
  - The distributions over $X$ and $K$ are denoted by the same $P$.
  - The line above Eq.(12), where $\mathcal{H}\ \subset \mathcal{Y}^{\mathcal{X}}$ is confusing.
  - $\omega^\star$ in Eq.(11) is inconsistent with the notation in the equation on Page 8.
  - Some equations are not numbered.

- Assuming the same true labeling functions $\mathcal{T}_S=\mathcal{T}_T$ is apparently not feasible for DA.

- Is the alignment achieved in feature space $g:\mathcal{X}\to \mathcal{Z}$? There is no such explanation throughout the paper.

- What is the cost of the diagonal covariance matrix assumption? Does it harm the performance?


- Euclidean distance is no longer discriminative due to the dimensionality curse. Therefore, I can hardly believe the proposal can work for an increasing label space.

- The details of training a source GMM and target GMM are not explained. Pseudo code describing the entire algorithm is encouraged for readability.

- The intention to assign multiple Gaussian components to each class for the source domain is unclear. E.g., mapping source data into Gaussian clusters can be easily achieved by a ResNet backbone with a Bayesian linear classifier.

- What is the difference between setting $\tau=0$ and previous work? How do we choose $\tau$ in practice? If a transport plan $\omega$ splitting $P_k$ into several $Q_k$ is discouraged, why not add some constraints to make it sparse?

- The author claims the contribution of generalization towards novel samples out of the source support. However, judging from the $T_{rand}$ on Page 8, I do not see why the existing estimation is only defined on empirical source data.


- The author claims the contribution on scalability, but the results on large-scale datasets, e.g., VisDA and DomainNet, are not provided. In addition, comparisons regarding the actual time consumption are not presented.

**Strengths And Weaknesses:**

Strengths:
- Optimal transport is a theoretically grounded tool for analyzing distribution changes.
- The result of $M\to S$ is impressive.

Weaknesses:
- I am not an expert in OT. However, most of the contents seem to come from previous works. Therefore, the novelty and contributions are not well elaborated regarding the scalability and generalization towards novel samples out of the source support.

- The experimental comparisons are only conducted within OT-based methods, far below SOTA, while non-OT baselines like FixBi [1] achieve much better results, e.g., 72.7\% in Office-Home and 91.4% in Office-31.

-  In visual DA tasks, the proposed method's advantage is marginal except for Caltech-Office 10, which only contains 10 classes. Therefore, I doubt this proposal can be generalized to larger label space cases.

[1] Fixbi: Bridging domain spaces for unsupervised domain adaptation, CVPR 2021

---

> ### Author Response · Authors · 2024-11-16
> **Rebuttal (1 / 3)**
>
> Dear Reviewer,
>
> Thank you for your feedback on our work. We are pleased that you found our work to be based on a theoretically grounded tool for analyzing distribution changes.
>
> __We would like to begin by emphasizing the novelty and contribution of our work.__ Our main contribution to domain adaptation are two methods based on the optimal transport of Gaussian mixture models. The first method, relies on label propagation between the components of source and target domain GMMs through a GMM-OT plan. The second method, relies on a new notion of mapping between GMMs that enforces that nearby samples are mapped together. Neither of these components are present in previous works.
>
> In addition, you raised a valid concern about our empirical results,
>
> > The experimental comparisons are only conducted within OT-based methods, far below SOTA, while non-OT baselines like FixBi [1] achieve much better results, e.g., 72.7% in Office-Home and 91.4% in Office-31.
>
> Please note that, contrary to our method, FixBi is a __deep__ domain adaptation method. This means that, throughout adaptation, it is capable of updating the encoder's parameter, changing the representation space of images. For visual domain adaptation tasks, this additional flexibility usually leads to better performance than shallow methods, such as ours and the other methods tested. However, it is difficult to compare these methods on a fair basis, as the number of variables in the optimization problems are quite different. As a result, we chose to keep our comparison on __shallow__, optimal transport-based methods.
>
> > The writing, especially regarding notations, requires an extensive revision.
>
> Thank you for pointing out our notation mistakes. We will correct those in the updated version of our manuscript. However, we would like to clarify the following point,
>
> > $\omega^{\star}$ in Eq.(11) is inconsistent with the notation in the equation on Page 8.
>
> We are not sure what the reviewer meant by inconsistent notation here. We would like to emphasize the difference between the OT plan $\gamma$, and the GMM-OT plan $\omega$.
>
> In GMM-OT, on one hand $\gamma$ is a continuous OT plan, which is constrained to be a GMM. On the other hand $\omega$ is a __discrete__ OT plan between components of the GMMs. $\gamma$ and $\omega$ are related through the formula (see Proposition 4 of Delon and Desoulneux, 2020).
>
> $\gamma(x\_{1}, x\_{2}) = \sum\_{k\_{1}=1}^{K\_{P}}\sum\_{k\_{2}=1}^{K\_{Q}}\omega\_{k\_{1},k\_{2}}\mathcal{N}(x\_{1}|\mu^{P}\_{k\_{1}},\Sigma\_{k\_{1}}^{(P)})\delta\_{y=T\_{k\_{1},k\_{2}}(x)}$
>
> where $T\_{k\_{1},k\_{2}}$ is the OT map between component $k\_{1}$ of $P$ and $k\_{2}$ of $Q$. While we choose not to go into the details of this relationship for the sake of concision, we can include more theoretical details into the final manuscript.
>
> > Assuming the same true labeling functions $\mathcal{T}\_{S} = \mathcal{T}\_{T}$  is apparently not feasible for DA.
>
> This assumption is common in works based on optimal transport maps (e.g., Courty et al., 2016). Could the reviewer give further details on the "non-feasibility" of this assumption?
>
> > Is the alignment achieved in feature space $g:\mathcal{X}\rightarrow\mathcal{Z}$? There is no such explanation throughout the paper.
>
> Our method aligns $P$ and $Q$ on feature space, that is, $\mathcal{Z}$. Note, however, that our method is a __shallow__ domain adaptation method, which means that we do not update the parameters $\theta_{g}$ of the encoder $g$.
>
> > What is the cost of the diagonal covariance matrix assumption? Does it harm the performance?
>
> Assuming diagonal covariance matrices reduces the representation power of the GMM model. As a result, this assumption requires us to use more components to express the same probability distribution. In another direction, one can consider that this choice leads to a regularizing effect, as we're estimating a simpler model with less parameters (diagonal covariances have $d$ parameters, while full covariances have $\dfrac{d(d-1)}{2}$). This approach strikes a balance between the computational and statistical complexity of estimating covariances, and the number of components in a GMM. In our experiments, we found that estimating full covariances in not feasible for most datasets.
>
> > The author claims the contribution of generalization towards novel samples out of the source support. However, judging from the $T_{rand}$ on Page 8, I do not see why the existing estimation is only defined on empirical source data.
>
> Could the reviewer clarify this point? For instance, the definition of $T_{rand}$ in page 8 does not rely on empirical data, but rather on the source domain density and the GMM-OT plan. For domain adaptation, however, we apply the map on the source domain data, to generate data on the target domain.

---

> > ### Comment · Reviewer_MxMx · 2024-11-21
> >
> > - Thank you for the detailed answers.
> >
> > - What is the advantage of this "shallow" method compared to "deep" ones?  I notice the proposed method has an impressive result in $M\to S$ from the digits dataset. A theoretical analysis of the reason for this interesting phenomenon would strengthen the claim.
> >
> > - Ben-David et al., 2010 take different labeling functions. The assumption for an identical labeling function is stronger than a small joint error $\lambda$. Considering the performance of ViSDA, it barely holds in a large domain shift.
> >
> > - So $P, Q$  are feature distributions given by  a not learnable $g$?

---

> > > ### Author Response · Authors · 2024-11-25
> > > **Response to Official Comment by Reviewer MxMx (2 / 2)**
> > >
> > > > Ben-David et al., 2010 take different labeling functions. The assumption for an identical labeling function is stronger than a small joint error $\lambda$. Considering the performance of ViSDA, it barely holds in a large domain shift.
> > >
> > > Indeed, there was a little confusion in our part as how we stated the assumptions for our method, and its relationship with the assumptions made in (Ben-David et al., 2010). We do not actually have the same ground-truth labeling function $h\_{S} = h\_{T}$. However, as (Courty et al., 2016), we do assume that these labeling functions are preserved under the action of the transformation $T$, that causes the distribution shift. As analyzed in (Redko et al., 2017), this is related to having a small joint error $\lambda$. In our revised manuscript, we removed the statement "and a single task $\mathcal{T}\_{S} = \mathcal{T}\_{T}$" for "and a single label space $\mathcal{Y}\_{S} = \mathcal{Y}\_{T}$".
> > >
> > > Concerning VisDA, we made additional experiments with other feature extractors, such as ResNet 50 and ViT-16-b, which showed promising results, especially with ViT,
> > >
> > > | Algorithm                   | ResNet 50 | ResNet101 | ViT-b-16  |
> > > |-----------------------------|-----------|-----------|-----------|
> > > | Source-Only                 | 47.93     | 53.90     | 56.70     |
> > > | OTDA$_{\text{EMD}}$         | 53.69     | 57.42     | 63.25     |
> > > | OTDA$_{\text{Sinkhorn}}$    | 53.02     | 10.54     | 66.75     |
> > > | OTDA$_{\text{Affine}}$      | 7.46      | 11.82     | 6.75      |
> > > | OTDA$_{\text{Affine-Diag}}$ | 51.41     | 56.91     | 59.94     |
> > > | HOTDA                       | 47.51     | 47.64     | 62.55     |
> > > | GMM-OTDA$_{\text{T}}$       | **58.35** | 56.57     | 74.10     |
> > > | GMM-OTDA$_{\text{MAP}}$     | 57.36     | **58.77** | **74.44** |
> > >
> > > Note that, in the case of ViT-b-16, we improve over the source-only baseline by a margin of 17.74\%.
> > >
> > > > So $P, Q$ are feature distributions given by a not learnable $g$?
> > >
> > > Yes.
> > >
> > >
> > > ## References
> > >
> > > Arthur P Dempster, Nan M Laird, and Donald B Rubin. Maximum likelihood from incomplete data via the em algorithm. Journal of the royal statistical society: series B (methodological), 39(1):1–22, 1977.
> > >
> > > Montesuma, E. F., Mulas, M., Mboula, F. N., Corona, F., & Souloumiac, A. (2024). Benchmarking Domain Adaptation for Chemical Processes on the Tennessee Eastman Process. In Machine Learning for Chemistry and Chemical Engineering (ML4CCE) Workshop.

---

> ### Author Response · Authors · 2024-11-16
> **Rebuttal (2 / 3)**
>
> > The details of training a source GMM and target GMM are not explained. Pseudo code describing the entire algorithm is encouraged for readability.
>
> Thank you for pointing out this problem. We will include pseudo codes for fitting the GMM models, and an entire pseudo code for our methods.
>
> > The intention to assign multiple Gaussian components to each class for the source domain is unclear. E.g., mapping source data into Gaussian clusters can be easily achieved by a ResNet backbone with a Bayesian linear classifier.
>
> Thank you for pointing that out. This remark actually confirms our idea of using GMMs for modeling the probability distributions. For instance, assuming that each class follows a Gaussian distribution implies that the overall distribution is a GMM. It is natural to assume that, in the general case, one might need Gaussian mixtures to model more complex distributions. We checked this assumption in our ablation in Appendix B.2, where we analyse performance on Image-CLEF as a function of number of components and entropic regularization. Note that, in most cases, using more than one Gaussian component per class leads to better performance.
>
> > What is the difference between setting $\tau = 0$ and previous work? How do we choose $\tau$ in practice? If a transport $\omega$ splitting $P\_{k}$ into several $Q\_{k}$  is discouraged, why not add some constraints to make it sparse?
>
> To the best of our knowledge, a parameter like $\tau$ was never considered before. We introduced this parameter to avoid creating too many unimportant maps $T\_{k\_{1},k\_{2}}(x^{(P)})$, for which they would receive a small importance factor $\omega\_{k\_{1},k\_{2}}$. For instance, if $x^{(P)}$ belongs to component $k\_{1}$ of $P$, and is mapped to all $Q\_{k\_{2}}$, the mapped points $T\_{k\_{1},k\_{2}}(x^{(P)})$ with $\omega\_{k\_{1},k\_{2}} = 0$ will naturally not be considered during training. To avoid making those irrelevant mappings, one can choose to filter out the entries $\omega\_{k\_{1},k\_{2}} = 0$.
>
> In our experiments, we choose $\tau$ as the quantile $\tau = quantile\_{\omega}(1 - \dfrac{K\_{P} + K\_{Q} - 1}{K_{P}K_{Q}})$, i.e., the value for which there are only $K\_{P} + K\_{Q} - 1$ entries bigger than $\tau$. Here, $quantile\_{\omega}$ corresponds to the quantile function over the flattened GMM-OT plan $\omega$.
>
> We will add those details in the revised version of our manuscript.

---

> > ### Comment · Reviewer_MxMx · 2024-11-21
> >
> > - Thank you for the explanation.
> > - I recommend the author include an algorithm during this discussion phase as TMLR allows authors to submit a revision.
> > - I am not sure if my understanding is correct. Still, the motivation for introducing multiple components seems to be not-learnable $\theta_g$, which cannot group the features from the same class into a single Gaussian.

---

> ### Author Response · Authors · 2024-11-16
> **Rebuttal (3 / 3)**
>
> > The author claims the contribution on scalability, but the results on large-scale datasets, e.g., VisDA and DomainNet, are not provided. In addition, comparisons regarding the actual time consumption are not presented.
>
> Following your request, as well as other reviewers, we include results on VisDA and a running time analysis of all methods considered.
>
> __Results on VisDA.__ We pre-train a ResNet-101 on the source domain data, then proceed to extract the source and target domain features for domain adaptation. Here are our results,
>
> | Algorithm | Accuracy on target domain |
> |-----------|---------------------------|
> |OTDA$\_{EMD}$|57.42|
> |OTDA$\_{Sinkhorn}$|10.54|
> |OTDA$\_{Affine}$|11.82|
> |HOTDA|47.64|
> |GMM-OTDA$\_{T}$|56.57|
> |GMM-OTDA$\_{MAP}$|58.77|
>
> It is important to clarify how OTDA and HOTDA were implemented on this benchmark. We subsample the source and target domain, selecting $n = 15000$ samples from each. This approach simplifies the adaptation problem, making it computationally feasible. The source domain data is then transported into the target domain, where it is used to train a classifier, and he trained classifier is subsequently used to classify all samples in the target domain. Performance is evaluated in terms of classification accuracy over the $n_{T} = 55388$ samples.
>
> Regarding the results in the table, OTDA$\_{sinkhorn}$ and OTDA$\_{Affine}$ perform significantly worse than the other methods. Specifically, the estimated covariances in the source and target domains are singular, making matrix inversion and square root calculations particularly challenging. Even by regularizing $\tilde{\Sigma}^{(P)} = \Sigma^{(P)} + \lambda I$ we could not get meaningful results.
>
> To demonstrate the effectiveness of axis-aligned Gaussian measures in such cases, we evaluated the affine map estimation using covariances of the form  $\Sigma = \text{diag}(\sigma\_{1},\cdots,\sigma\_{d})$, which leads to the map,
>
> $$T(\mathbf{x}) = \mathbf{Ax} + \mathbf{b}$$
>
> where $\mathbf{A} = \text{diag}(\sigma^{(Q)} / \sigma^{(P)})$ and $\mathbf{b} = \mu^{(Q)} - \mathbf{A}\mu^{(P)}$
>
> The axis-aligned covariance model achieved a significantly higher classification accuracy of $56.91\%$ compared to $11.82\%$ obtained using the affine map with full covariances.
>
> Overall, this experiment highlights two key observations:
>
> 1. Subsampling negatively impacts performance. Consequently, methods that efficiently represent probability measures, such as those using clustering, can better leverage a larger number of samples.
> 2. Axis-aligned covariance matrices provide a regularizing effect, improving the estimation of an OT map between Gaussian measures when the full-covariances are singular.
>
> __Running-Time Analysis.__ We performed a run-time analysis of our algorithm in comparison with the other tested algorithms with respect the adaptation task $A \rightarrow W$ on Office 31. In this case, $n\_{S} = 2817$, $n\_{T}=624$, $n\_{c} = 31$ and $d = 2048$. We ran each algorithm 10 independent times, then computed the mean and standard deviation of their running time.
>
>
> | Algorithm | Running Time (seconds) |
> |-----------|------------------------|
> |OTDA$\_{EMD}$|0.775 $\pm$ 0.007     |
> |OTDA$\_{Sink}$|14.119 $\pm$ 0.125   |
> |OTDA$\_{Affine}$|8.503 $\pm$ 0.097  |
> |InfoOT| 105.407 $\pm$ 0.716 	     |
> |HOTDA|2.508 $\pm$ 0.023 	     |
>
> Now, concerning our GMM-OTDA, the complexity depends on $K$. We report our running time with respect $K = 31, 62, 93, 124, 155, 186, 217$,
>
> | Number of components | Running Time (seconds) |
> |----------------------|------------------------|
> |31|   0.742 $\pm$ 0.074 |
> |62|   1.329 $\pm$ 0.015 |
> |93|   1.995 $\pm$ 0.035 |
> |124|  2.679 $\pm$ 0.051 |
> |155|  3.517 $\pm$ 0.025 |
> |186|  4.343 $\pm$ 0.090 |
> |217|  5.069 $\pm$ 0.059 |
>
> Overall, notice how our method is _faster_ than most of the previous methods, with the exception of OTDA$\_{EMD}$ and $HOTDA$. These results highlight our claim that our method improves the scalability of OT-based domain adaptation methods, mainly due to 2 factors,
>
> - The OT problem being solved is $K\_{P} \times K\_{Q}$ (number of components by number of components). Since $K\_{P}$ and $K\_{Q}$ are ideally much smaller than $n\_{S}$ and $n\_{T}$, we have a lower computational complexity while solving OT.
> - Furthermore, we alleviate the storage complexity of OTDA, which needs to store a $n\_{S} \times n\_{T}$ transport plan. For large datasets (such as MNIST, SVHN, or VisDA) this is prohibitive.

---

> > ### Comment · Reviewer_MxMx · 2024-11-21
> >
> > Thank you for the elaboration and additional results.

---

> ### Author Response · Authors · 2024-11-25
> **Response to Official Comment by Reviewer MxMx (1 / 2)**
>
> Dear Reviewer,
>
> Thank you for your feedback. Here are our answers to your questions,
>
> > I recommend the author include an algorithm during this discussion phase as TMLR allows authors to submit a revision.
>
> We added 4 algorithms to the main paper, summarizing the different components of our methods. Algorithm 1 shows the standard Expectation-Maximization Algorithm as proposed by (Dempster et al., 1977). Algorithm 2 shows our strategy for fitting a GMM to each $P_{y} = P(X|Y=y)$. Algorithm 3 shows our label propagation strategy. Algorithm 4 shows how to compute $T_{weight}$.
>
> > I am not sure if my understanding is correct. Still, the motivation for introducing multiple components seems to be not-learnable $\theta_{g}$, which cannot group the features from the same class into a single Gaussian.
>
> The motivation for multiple components __is not__ having a not-learnable $\theta_{g}$. Here are some comments in this regard,
>
> 1. Our method has a different focus from deep methods. On the one hand, these methods seek for $\theta_{g}$ such that domains are aligned in the latent space (i.e., $\mathcal{Z}$). These methods tend to learn __domain invariant features__, which may, or may not be discriminative with respect the classes. Our approach, on the other hand, seeks to map latent representations from the source domain towards the target domain, so that training can be carrier directly on the target domain. Arguably, one could perform the two tasks at once, but this would require non-trivial changes to our setting, especially concerning the fit of GMMs while changing the representations. We do consider this possibility for future works.
> 2. We choose to model the overall $P(X, Y)$ through a GMM. We do so by modeling the class conditionals $P_{y} = P(X|Y=y)$ through GMMs, then concatenating the components altogether. This way, we can assign labels to the components of the GMM representing $P(X, Y)$, which are later used for labeling the target GMM, which models $Q(X)$.
>
> Concerning the latter statement "_which cannot group the features from the same class into a single Gaussian_",
>
> 3. Our framework can handle cases where $P(X|Y=y)$ is a single Gaussian, as it suffices to set _components per class_ as 1.
> 4. More generally, using GMMs instead of single Gaussians improve performance (see, e.g., our ablations in Appendix B). We also refer the reviewer to the t-SNE visualizations of VisDA and CWRU in the revised version of our paper, as these show clusters that are potentially more complex than Gaussian distributions.
>
> > What is the advantage of this "shallow" method compared to "deep" ones? I notice the proposed method has an impressive result in $M \rightarrow S$ from the digits dataset. A theoretical analysis of the reason for this interesting phenomenon would strengthen the claim.
>
> Shallow DA methods offer an advantage as being lightweight with respect deep ones. The fact that we do not backpropagate through the encoder's parameter alleviates the computational burden necessary to perform DA. In addition to this, in cases where a general purpose, pre-trained deep neural net (such as ResNets) is not available (e.g., time series classification, such as fault diagnosis), shallow DA methods oftentimes have a more reliable performance. This is further discussed in the experiments of (Montesuma et al., 2024).
>
> Now, concerning the superiority of our method in the $M \rightarrow S$ case, except for LaOT, note that the the other tested methods __are not__ deep learning based methdos, i.e., they do not update the encoder network either. The superiority of our method in comparison with Seguy et al., (2017) may stem from the fact that these authors employ batch optimization for the neural nets representing the dual variables in OT. This has been studied, for instance, by Fatras et al. (2019) to be problematic if the batch size is not big enough. Overall, the result indicate that through GMM-OT do a better job at estimating large scale OT in this particular adaptation task.
>
> Now, for LaOT, the situation is more complex, since these authors try to learn a latent space where one can map between domains with an affine map. However, in classification, source and target distributions are still multi-modal, which means that representing these dsitributions through Gaussians is likely innacurate.
>
> ## References
>
> Fatras, K., Zine, Y., Flamary, R., Gribonval, R., & Courty, N. (2019). Learning with minibatch Wasserstein: asymptotic and gradient properties. arXiv preprint arXiv:1910.04091.

---

> > ### Comment · Reviewer_MxMx · 2024-11-28
> >
> > - Thank you for the elaboration, which addresses some of my questions.
> > - However, the explanation regarding the advantage is still not convincing.
> > - The "computational burden" hardly makes sense, as backpropagation in DA is not a heavy task for GPUs.
> > - The author mentioned time series classification, but the corresponding experiments are not included in this manuscript.
> > - To my knowledge, the result in M$\to$S is quite impressive even compared to deep methods. That is why I suggest the author consider a further analysis, e.g., the circumstances where shallow methods can outperform deep ones.
> > - It can be subjective, but the importance of computational efficiency compared to other OT baselines seems marginal when the accuracy of the method is far below SOTA. In VisDA, even a source-free method (Liang et al., 2020) can achieve 87\%, while the proposed method can only obtain 59\%, which makes it hard to assess the contribution of this work.
> >
> >
> >
> > Source Data-absent Unsupervised Domain Adaptation through Hypothesis Transfer and Labeling Transfer, Liang et al., 2020

---

> > > ### Author Response · Authors · 2024-11-30
> > > **Response to Official Comment by Reviewer MxMx**
> > >
> > > Dear Reviewer,
> > >
> > > Thank you for your feedback. To begin, we would like to emphasize a key point in our work.
> > >
> > > Our proposed methods __align__ source and target domains with respect their distributions. This alignment is done at the level of the latent space of a neural net, i.e., over features $\set{g(\mathbf{x}\_{i}^{(P_{S})})}\_{i=1}^{n}$ and $\set{g(\mathbf{x}\_{j}^{(P_{T})})}\_{j=1}^{m}$.
> > >
> > > This strategy is different from deep DA methods, which try to align the domains by learning domain-invariant features, i.e., the encoder network $g$ is updated throughout learning so that source and target domains are indistinguishable. Note that, in principle, the methods compared in our paper can be plugged after domain-invariant feature learning.
> > >
> > > We agree with the reviewer’s observation that domain-invariant learning strategies generally achieve better performance for visual domain adaptation tasks. As we mentioned before, a comparison with such methods is out of scope of our paper, since a fair comparison is not possible. If the reviewer finds it valuable, we can include a discussion about these considerations in our manuscript.
> > >
> > > Furthermore, we stress that __our contribution is still valid__, since,
> > >
> > > - Our methods consistently improve the source-only baseline, over a variety of feature extractors.
> > > - Our methods consistently outperform previous OT-based mapping techniques.
> > > - Our performance is indeed below __deep__ SOTA, but, as we mentioned, the comparison is not fair.
> > >
> > > Second, here are our answers to the points you mentioned,
> > >
> > > > The "computational burden" hardly makes sense, as backpropagation in DA is not a heavy task for GPUs.
> > >
> > > When referring to _computational burden_, we meant that updating the encoder's network $\theta_{g}$ has a higher computational cost. For instance, in the case of a ResNet 101 or Vit-16-b, training the whole network on GPU with a batch-size of 32 samples requires over 8GB of GPU. If samples from the target domain are passed through the network as well, this requirement is increased. In contrast, only retraining the classification head can even be done on CPU. For applications where hardware is limited (e.g., embedded systems), performing deep DA is not possible.
> > >
> > > > The author mentioned time series classification, but the corresponding experiments are not included in this manuscript.
> > >
> > > We do include experiments with time series. In the original version of our manuscript, these were shown in the experiments section over Cross-Domain Fault Diagnosis. We moved this section to the appendix (Appendix B.3), which figures experiments with the CWRU, CSTR and TEP benchmarks. All of these benchmarks deal with classification tasks over time series.
> > >
> > > We did not include the comparisons of (Montesuma et al., 2024) with deep models since these are out of the scope of the current paper.
> > >
> > > > It can be subjective, but the importance of computational efficiency compared to other OT baselines seems marginal when the accuracy of the method is far below SOTA. In VisDA, even a source-free method (Liang et al., 2020) can achieve 87%, while the proposed method can only obtain 59%, which makes it hard to assess the contribution of this work.
> > >
> > > While we obtain 58.77\% with ResNet-101 features, we highlight that we achieve over 74.44\% on ViT-b-16, that is, an improvement of 17.74\% over the source-only performance. While the performance is inferior to what deep methods report, our experiments show that __we do improve over the baseline across different archiectures__, and __we outperform previous OT-based methods__.

---

> > > > ### Comment · Reviewer_MxMx · 2024-11-30
> > > >
> > > > Thank you for the answers. I see the results of fault diagnosis, but the comparisons with deep methods are not included, which is insufficient to support the author's claim that shallow methods often have more reliable performance in time series. Once I have reviewed the comparisons with reasonable explanations, I can recommend the paper for acceptance.

---

> > > > > ### Author Response · Authors · 2024-12-03
> > > > > **Response to Official Comment by Reviewer MxMx**
> > > > >
> > > > > Dear Reviewer,
> > > > >
> > > > > Thank you for your comment, and your engagement during the review process. We followed your suggestion, and included further experiments with other deep methods in the context of the Tennessee Eastman Process benchmark. Here, we evaluate the advantage of shallow methods over deep methods. You may find the overall changes in our main paper in Appendix B.3, highlighted in blue. We summarize our findings in the following table,
> > > > >
> > > > > | Method | Average adaptation performance |
> > > > > |-----------|----------------------------------------------|
> > > > > | Baseline| 48.01|
> > > > > |OTDA$\_{\text{EMD}}$| 60.37 |
> > > > > |OTDA$\_{\text{Sinkhorn}}$ | 59.39 |
> > > > > |OTDA$\_{\text{Affine}}$| 56.76 |
> > > > > |HOTDA|57.34|
> > > > > |GMM-OTDA$\_{\text{MAP}}$|59.72|
> > > > > |GMM-OTDA$\_{\text{T}}$|59.48|
> > > > > |DANN|36.11|
> > > > > |DAN|51.28|
> > > > > |WDGRL|48.21|
> > > > > |DeepJDOT|57.88|
> > > > >
> > > > > Where DANN (Ganin et al., 2016), DAN (Ghifary et al., 2014), WDGRL (Shen et al., 2018) and DeepJDOT (Damodaran et al., 2018) are methods based on learning domain invariant features. We did experiment with other methods suggested by reviewers during this review process, such as FixBi (suggested by you), CoVi, and MARS (Suggested by reviewer __hmXt__), but we could not obtain meaningful results.
> > > > >
> > > > > ## References
> > > > >
> > > > > Yaroslav Ganin, Evgeniya Ustinova, Hana Ajakan, Pascal Germain, Hugo Larochelle, François Laviolette,
> > > > > Mario March, and Victor Lempitsky. Domain-adversarial training of neural networks. Journal of machine
> > > > > learning research, 17(59):1–35, 2016.
> > > > >
> > > > > Muhammad Ghifary, W Bastiaan Kleijn, and Mengjie Zhang. Domain adaptive neural networks for object
> > > > > recognition. In PRICAI 2014: Trends in Artificial Intelligence: 13th Pacific Rim International Conference
> > > > > on Artificial Intelligence, Gold Coast, QLD, Australia, December 1-5, 2014. Proceedings 13, pp. 898–904.
> > > > > Springer, 2014
> > > > >
> > > > > Jian Shen, Yanru Qu, Weinan Zhang, and Yong Yu. Wasserstein distance guided representation learning for
> > > > > domain adaptation. In Proceedings of the AAAI conference on artificial intelligence, volume 32, 2018
> > > > >
> > > > > Bharath Bhushan Damodaran, Benjamin Kellenberger, Rémi Flamary, Devis Tuia, and Nicolas Courty.
> > > > > Deepjdot: Deep joint distribution optimal transport for unsupervised domain adaptation. In Proceedings
> > > > > of the European conference on computer vision (ECCV), pp. 447–463, 2018.

---

> > > > > > ### Comment · Reviewer_MxMx · 2024-12-04
> > > > > >
> > > > > > I appreciate the author's effort during this discussion phase, which has addressed most of my concerns. Therefore, I recommend accepting the paper for publication.

---

> > > > > > > ### Author Response · Authors · 2024-12-04
> > > > > > > **Response to Official Comment by Reviewer MxMx**
> > > > > > >
> > > > > > > Dear reviewer,
> > > > > > >
> > > > > > > We are glad that our answers addressed most of your concerns. Thank you for your valuable feedback, which helped make our submission stronger.

---

### Review · Reviewer_hmXt · 2024-11-08

**Summary Of Contributions:**

This paper builds on the GMMOT framework, applying it to domain adaptation and proposing two strategies. Compared to previous approaches, the method reduces time complexity and demonstrates performance improvements on multiple benchmarks.

**Audience:**

Yes

**Broader Impact Concerns:**

No concerns.

**Claims And Evidence:**

Yes

**Requested Changes:**

- Please consider comparing your method with the following OT-based works [r1, r2] and other state-of-the-art UDA approaches. To my knowledge, CoVi [r3] is a relevant option.

  [r1] Optimal transport for conditional domain matching and label shift, Machine Learning 2021.

  [r2] Mapping conditional distributions for domain adaptation under generalized target shift, ICLR 2022.

  [r3] Contrastive Vicinal Space for Unsupervised Domain Adaptation, ECCV 2022.

- In the paper, the authors mention that proposed method has a smaller computational complexity and that InfoOT does not exhibit reasonable runtime. Could you provide specific runtime comparisons for clarity?

**Strengths And Weaknesses:**

Summary of strengths

- The paper is well-organized, clear and easy to follow.
- The proposed GMM-OTDA method demonstrates better scalability than previous OT-based domain adaptation methods.
- Theoretical analysis is provided for the proposed approach.

Summary of weaknesses

- The method appears to have limited scalability on larger datasets, as demonstrated in Table 5. Specifically, it does not achieve optimal performance on datasets with a large number of classes, such as Office-31 and Office-Home. More complex and widely used UDA datasets like VisDA and DomainNet were not addressed.
- The related work section is somewhat limited. Although the paper claims state-of-the-art performance, it only compares against a few OT-based UDA methods, lacking a more comprehensive evaluation.
- The practical contribution of optimal transport for domain adaptation in this work is limited for complex datasets, as it relies heavily on feature extraction from pretrained models. Additionally, when classes are imbalanced, the effectiveness of optimal transport can be compromised.

- Some unclear points:
  - In Table 1, further introduction of the compared methods would enhance clarity.
  - In the Baselines and Backbones section, $GMM-OTDA_{MAP}$ is described, but there appears to be insufficient detail on the other proposed method, $GMM-OTDA_{T}$.

---

> ### Author Response · Authors · 2024-11-16
> **Rebuttal (1 / 3)**
>
> Dear reviewer,
>
> Thank you for your feedback on our work. We are pleased that you acknowledged that our work is __well organized, clear and easy to follow__, and that our method demonstrates __better scalability__ than previous OT-based approaches. We will address the points you raised regarding the weaknesses, particularly the descriptions of methods in Table 1 and the details of GMM-OTDA$_{T}$.
>
> In this official comment, we would like to answer your list of requested changes.
>
> ## On considering other benchmarks
>
> > The method appears to have limited scalability on larger datasets, as demonstrated in Table 5. Specifically, it does not achieve optimal performance on datasets with a large number of classes, such as Office-31 and Office-Home. More complex and widely used UDA datasets like VisDA and DomainNet were not addressed.
>
> Thank you for your suggestion. We added experiments on the VisDA benchmark. We pre-train a ResNet-101 on the source domain data, then proceed to extract the source and target domain features for domain adaptation. Here are our results,
>
> | Algorithm | Accuracy on target domain |
> |-----------|---------------------------|
> |OTDA$\_{EMD}$|57.42|
> |OTDA$\_{Sinkhorn}$|10.54|
> |OTDA$\_{Affine}$|11.82|
> |HOTDA|47.64|
> |GMM-OTDA$\_{T}$|56.57|
> |GMM-OTDA$\_{MAP}$|58.77|
>
> It is important to clarify how OTDA and HOTDA were implemented on this benchmark. We subsample the source and target domain, selecting $n = 15000$ samples from each. This approach simplifies the adaptation problem, making it computationally feasible. The source domain data is then transported into the target domain, where it is used to train a classifier, and he trained classifier is subsequently used to classify all samples in the target domain. Performance is evaluated in terms of classification accuracy over the $n_{T} = 55388$ samples.
>
> Regarding the results in the table, OTDA$\_{sinkhorn}$ and OTDA$\_{Affine}$ perform significantly worse than the other methods. Specifically, the estimated covariances in the source and target domains are singular, making matrix inversion and square root calculations particularly challenging. Even by regularizing $\tilde{\Sigma}^{(P)} = \Sigma^{(P)} + \lambda I$ we could not get meaningful results.
>
> To demonstrate the effectiveness of axis-aligned Gaussian measures in such cases, we evaluated the affine map estimation using covariances of the form  $\Sigma = \text{diag}(\sigma\_{1},\cdots,\sigma\_{d})$, which leads to the map,
>
> $$T(\mathbf{x}) = \mathbf{Ax} + \mathbf{b}$$
>
> where $\mathbf{A} = \text{diag}(\sigma^{(Q)} / \sigma^{(P)})$ and $\mathbf{b} = \mu^{(Q)} - \mathbf{A}\mu^{(P)}$
>
> The axis-aligned covariance model achieved a significantly higher classification accuracy of $56.91\%$ compared to $11.82\%$ obtained using the affine map with full covariances.
>
> Overall, this experiment highlights two key observations:
>
> 1. Subsampling negatively impacts performance. Consequently, methods that efficiently represent probability measures, such as those using clustering, can better leverage a larger number of samples.
> 2. Axis-aligned covariance matrices provide a regularizing effect, improving the estimation of an OT map between Gaussian measures when the full-covariances are singular.

---

> > ### Comment · Reviewer_hmXt · 2024-11-22
> >
> > Thank you for your explanation. I would like to ask how the ResNet-101 was pre-trained. Did you use an existing method for pre-training? What was the initial accuracy of the pre-trained model? Furthermore, I am curious to know whether GMM-OTDA consistently improves domain adaptation performance across different feature extractors.
> >
> > Thank you for your clarification and insights.

---

> > > ### Author Response · Authors · 2024-11-25
> > > **Response to Official Comment by Reviewer hmXt (1 / 2)**
> > >
> > > These are really interesting questions. Here are our answers,
> > >
> > > > Could you also provide the corresponding accuracies for these algorithms?
> > >
> > > We appended a column corresponding to the accuracy, which helps giving a better picture at how these two values are related,
> > >
> > > | Algorithm | Running Time (seconds) | Accuracy |
> > > |-----------|------------------------|---------------------------|
> > > |OTDA$\_{EMD}$|0.775 $\pm$ 0.007     |  74.27  |
> > > |OTDA$\_{Sink}$|14.119 $\pm$ 0.125   | 80.12 |
> > > |OTDA$\_{Affine}$|8.503 $\pm$ 0.097  | 80.12 |
> > > |InfoOT| 105.407 $\pm$ 0.716 	     | 79.53 |
> > > |HOTDA|2.508 $\pm$ 0.023 	     |  73.68  |
> > >
> > > and the same for GMM-OTDA,
> > >
> > > | Number of Components | Running Time (seconds) | GMM-OTDA$_{\text{T}}$ | GMM-OTDA$_{\text{MAP}}$ |
> > > |-----------------------|------------------------|------------------------|--------------------------|
> > > | 31                   | 0.742 $\pm$ 0.074     | 78.94                 | 64.91                   |
> > > | 62                   | 1.329 $\pm$ 0.015     | 79.53                 | 70.76                   |
> > > | 93                   | 1.995 $\pm$ 0.035     | 76.02                 | 70.17                   |
> > > | 124                  | 2.679 $\pm$ 0.051     | 80.70                 | 74.85                   |
> > > | 155                  | 3.517 $\pm$ 0.025     | 78.36                 | 75.44                   |
> > > | 186                  | 4.343 $\pm$ 0.090     | 80.70                 | 70.17                   |
> > > | 217                  | 5.069 $\pm$ 0.059     | 77.77                 | 76.02                   |
> > >
> > > We added these results to our __Appendix B.4__

---

> > > ### Author Response · Authors · 2024-11-25
> > > **Response to Official Comment by Reviewer hmXt (2 / 2)**
> > >
> > > > It appears that OTDA$_{EMD}$ has a similar accuracy and running time to GMM-OTDA. Given this, would there be a significant difference in running time for these methods on the VisDA dataset?
> > >
> > > Note that OTDA$_{EMD}$ was run on $n = 15000$ samples selected at random from VisDA, whereas GMM-OTDA runs on the complete dataset. Here's a comparison over the VisDA dataset,
> > >
> > > | Algorithm                     | Running Time (seconds) |
> > > |-------------------------------|------------------------|
> > > |OTDA$\_{EMD}$ ($15000$ samples)| 33.10 $\pm$ 0.17       |
> > > |GMM-OTDA ($15000$ samples)     | 21.30 $\pm$ 0.15       |
> > >
> > > >  I would like to ask how the ResNet-101 was pre-trained. Did you use an existing method for pre-training? What was the initial accuracy of the pre-trained model?
> > >
> > > Following standard practice in domain adaptation, we pre-train the ResNet-101 model using source domain data only. Assuming a neural net $f = h \circ g$, where $g$ is the encoder network, and $h$ is the classifier, we tune $\theta_{g}$ and $\theta_{h}$ to minimize the cross-entropy loss,
> > >
> > > $\mathcal{L}(\mathbf{x}, \mathbf{y}) = -\sum_{c=1}^{n_{c}}y_{c}\log h(g(\mathbf{x}))$
> > >
> > > Here, we initialize $\theta_{g}$ from a pre-training on ImageNet (ImageNet1k\_V2 weights from torchvision, as you might check on torchvision documentation, https://pytorch.org/vision/stable/models.html).
> > >
> > > The source pre-training stage is carried out through 1200 iterations (12000 mini-batches of size 64), with an Adam optimizer with learning rate $10^{-5}$.
> > >
> > > > Furthermore, I am curious to know whether GMM-OTDA consistently improves domain adaptation performance across different feature extractors.
> > >
> > > We did an additional experiment to verify your question. We compared the performance of all tested methods over the following architectures: ResNet 50, 101 and Vit-b-16 (Dosovitskiy et al., 2021). All of these encoders are pretrained with the previously described strategy. We report the obtained accuracy on the table below,
> > >
> > > | Algorithm                   | ResNet 50 | ResNet101 | ViT-b-16  |
> > > |-----------------------------|-----------|-----------|-----------|
> > > | Source-Only                 | 47.93     | 53.90     | 56.70     |
> > > | OTDA$_{\text{EMD}}$         | 53.69     | 57.42     | 63.25     |
> > > | OTDA$_{\text{Sinkhorn}}$    | 53.02     | 10.54     | 66.75     |
> > > | OTDA$_{\text{Affine}}$      | 7.46      | 11.82     | 6.75      |
> > > | OTDA$_{\text{Affine-Diag}}$ | 51.41     | 56.91     | 59.94     |
> > > | HOTDA                       | 47.51     | 47.64     | 62.55     |
> > > | GMM-OTDA$_{\text{T}}$       | **58.35** | 56.57     | 74.10     |
> > > | GMM-OTDA$_{\text{MAP}}$     | 57.36     | **58.77** | **74.44** |
> > >
> > > We highlight a few findings from this experiment,
> > >
> > > - Using diagonal covariance matrices consistently outperforms the full covariance setting, which underlines the regularizing effect of using a simpler covariance structure. We stress that, besides the regularizing effect, estimating diagonal covariances is numerically more stable than estimating full covariance matrices (which are singular for the 3 tested feature extractors).
> > > - __GMM-OTDA consistently improves DA performance across different feature extractors.__ Furthermore, __we consistently outperform the empirical baseline__, OTDA$_{\text{EMD}}$.
> > >
> > > These highlights are described in more detail, with t-SNE visualizations, in __Section 5.3__ of our revised manuscript.
> > >
> > > ## References
> > >
> > > Alexey Dosovitskiy, Lucas Beyer, Alexander Kolesnikov, Dirk Weissenborn, Xiaohua Zhai, Thomas Unterthiner, Mostafa Dehghani, Matthias Minderer, Georg Heigold, Sylvain Gelly, Jakob Uszkoreit, and Neil Houlsby. An image is worth 16x16 words: Transformers for image recognition at scale. In International Conference on Learning Representations, 2021. URL https://openreview.net/forum?id=YicbFdNTTy.

---

> > > > ### Comment · Reviewer_hmXt · 2024-11-28
> > > >
> > > > Thanks to the author's response, which addressed most of my concerns. I have no further questions and recommend accepting the paper for publication.

---

> > > > > ### Author Response · Authors · 2024-11-30
> > > > >
> > > > > Dear reviewer,
> > > > >
> > > > > We are glad that our answers addressed most of your concerns. Thank you for your valuable feedback, which helped make our submission stronger.

---

> ### Author Response · Authors · 2024-11-16
> **Rebuttal (2 / 3)**
>
> ## On considering other baselines
>
> > Please consider comparing your method with the following OT-based works [r1, r2] and other state-of-the-art UDA approaches. To my knowledge, CoVi [r3] is a relevant option.
>
> Thank you for suggesting additional relevant references for our work.
>
> As with CWRU, InfoOT did not have a reasonable running time for this adaptation task. Note that GMM-OTDA$\_{MAP}$ is the best performing method among the tested ones. Furthermore, we significantly outperform both OTDA$\_{Affine}$ and OTDA$\_{Sinkhorn}$. In the first case, the estimation
>
>
> Although we would be glad to include comparisons to other methods, the suggested methods are somewhat outside the scope of our approach. Here are a few reasons on why comparisons with references [r1, r2] and [r3] would not be entirely fair,
>
> - Those 3 methods are __deep__ domain adaptation methods, which means that they are able to change the parameters of the encoder network during the adaptation process. This flexibility gives deep methods a particular advantage in visual domain adaptation benchmarks over __shallow__ methods (such as ours), which use fixed, pre-extracted features.
>
> - Beyond the fairness issues behind comparisons of deep and shallow methods, references [r1, r2] and [r3] have different training settings. For instance, [r1] and [r2] ran experiments on Office31 and Office-Home using a ResNet50 with a multi-layer perceptron on top of the feature extractor, while we use a single-layer Perceptron. This setting explains why these authors have an average performance of 47.86 for MARSc [r1] and 59.50 for OSTAR+IM [r2] - both inferior to our results (e.g., 67.40 for GMM-OTDA$_{T}$). In the case of [r3], after carefully reading the experimental settings used for training the ResNet, we noted that the authors used different transformations on the raw images, which impact the training and the generalization performances on top of the additional flexibility of being able to modify the encoder's parameters.
>
> Overall, while these references are indeed valuable, fair comparisons with our proposed method are not feasible, as these methods rely on fundamentally different principles.

---

> ### Author Response · Authors · 2024-11-16
> **Rebuttal (3 / 3)**
>
> ## On running time analysis
>
> > In the paper, the authors mention that proposed method has a smaller computational complexity and that InfoOT does not exhibit reasonable runtime. Could you provide specific runtime comparisons for clarity?
>
> Thank you for your suggestion. We performed a run-time analysis of our algorithm in comparison with the other tested algorithms with respect the adaptation task $A \rightarrow W$ on Office 31. In this case, $n\_{S} = 2817$, $n\_{T}=624$, $n\_{c} = 31$ and $d = 2048$. We ran each algorithm 10 independent times, then computed the mean and standard deviation of their running time.
>
>
> | Algorithm | Running Time (seconds) |
> |-----------|------------------------|
> |OTDA$\_{EMD}$|0.775 $\pm$ 0.007     |
> |OTDA$\_{Sink}$|14.119 $\pm$ 0.125   |
> |OTDA$\_{Affine}$|8.503 $\pm$ 0.097  |
> |InfoOT| 105.407 $\pm$ 0.716 	     |
> |HOTDA|2.508 $\pm$ 0.023 	     |
>
> Now, concerning our GMM-OTDA, the complexity depends on $K$. We report our running time with respect $K = 31, 62, 93, 124, 155, 186, 217$,
>
> | Number of components | Running Time (seconds) |
> |----------------------|------------------------|
> |31|   0.742 $\pm$ 0.074 |
> |62|   1.329 $\pm$ 0.015 |
> |93|   1.995 $\pm$ 0.035 |
> |124|  2.679 $\pm$ 0.051 |
> |155|  3.517 $\pm$ 0.025 |
> |186|  4.343 $\pm$ 0.090 |
> |217|  5.069 $\pm$ 0.059 |
>
> Overall, notice how our method is _faster_ than most of the previous methods, with the exception of OTDA$\_{EMD}$ and $HOTDA$. These results highlight our claim that our method improves the scalability of OT-based domain adaptation methods, mainly due to 2 factors,
>
> - The OT problem being solved is $K\_{P} \times K\_{Q}$ (number of components by number of components). Since $K\_{P}$ and $K\_{Q}$ are ideally much smaller than $n\_{S}$ and $n\_{T}$, we have a lower computational complexity while solving OT.
> - Furthermore, we alleviate the storage complexity of OTDA, which needs to store a $n\_{S} \times n\_{T}$ transport plan. For large datasets (such as MNIST, SVHN, or VisDA) this is prohibitive.

---

> > ### Comment · Reviewer_hmXt · 2024-11-22
> >
> > Thank you for your response on the running time analysis.
> >
> > Could you also provide the corresponding accuracies for these algorithms? It appears that OTDA$_{EMD}$ has a similar accuracy and running time to GMM-OTDA. Given this, would there be a significant difference in running time for these methods on the VisDA dataset?

---

### Author Response · Authors · 2024-11-16
**Rebuttal summary**

Dear reviewers,

We would like to thank all of you for your valuable feedback. In this official comment, we would like to summarize our rebuttal into a few key points that were raised in most reviews.

__Lack of experiments on more complex benchmarks.__ This point was raised by all reviewers. As a result, we added a comparison on the VisDA benchmark, which showed that our GMM-OTDA outperforms previous OT-based methods.

__Running-time analysis.__ This point was raised by all reviewers as well. We added a running-time analysis on the Office 31 benchmark, which shows that our method is faster than most tested methods. We also studied how the number of components on the GMMs affect the running time.

__On the consideration of other baselines.__ Reviewers __hmXt__ and __MxMx__ suggested that we should consider other methods in our comparison. While we would be glad to include other algorithms in our empirical validation, a comparison between shallow and deep domain adaptation methods is not possible, since these kinds of methods rely on different principles. As a result, we could not consider deep DA methods CoVi (recommended by Reviewer __hmXt__ ) nor FixBI (recommended by Reviewer __MxMx__).

Overall, we hope that our answers helped clarify the unclear points, as well as strong our empirical validation and arguments.

---

### Author Response · Authors · 2024-11-25
**Summary of changes in the manuscript**

Dear reviewers,

Thank you again for your valuable feedback. We just posted a revised version of our manuscript. Here is an overview of the changes,

- Following the comment of reviewer __MxMx__, we reviewed our notation. Here are a few highlights,
	- We now use $P_{S}$ and $P_{T}$ to denote source and target domain distribution. Uppercase letters such as $P$ and $Q$ are still used for OT theory.
$P_{S}$ and $P_{T}$ are used in the context of domain adaptation. For probabilities in general (e.g., conditionals of components $K$ given features $X$), we use $Pr$ (e.g., $Pr(K=k|X=\mathbf{x})$). We further added a paragraph on notation at the introduction of our paper.
- Following the commnets of reviewers __MxMx__, and __zrmE__ after rebuttal, we included pseudo-codes for the components of our method.
	- Algorithm 1 details the Expectation Maximization algorithm.
	- Algorithm 2 details the fitting procedure for the labeled GMMs, i.e., fitting a GMM on each conditional $P(X|Y=y)$ through an EM
	- Algorithm 3 details the label propagation of source GMM labels towards the target domain.
	- Algorithm 4 details the mapping procedure $T_{weight}$.
- In the spirit of adding further references on GMMs, we added a citation to (M. Bishop, 2006; Chapter 9) for readers that may not be familiar with the fundamentals of GMMs.
- With the intent to keep our main paper to 12 pages, and to make place for the algorithms, and experiments with VisDA-c, we moved the experiments with cross-domain fault diagnosis to the appendix. We also added a fourth section in the additional experiments part of the appendix, involving running time analysis.

Concerning our experiments,

- By the request of reviewer __hmXt__, we added new results on VisDA benchmark, which include other feature extractors such as ResNet50 and ViT-16-b. Here are the compiled results,

| Algorithm                   | ResNet 50 | ResNet101 | ViT-b-16  |
|-----------------------------|-----------|-----------|-----------|
| Source-Only                 | 47.93     | 53.90     | 56.70     |
| OTDA$_{\text{EMD}}$         | 53.69     | 57.42     | 63.25     |
| OTDA$_{\text{Sinkhorn}}$    | 53.02     | 10.54     | 66.75     |
| OTDA$_{\text{Affine}}$      | 7.46      | 11.82     | 6.75      |
| OTDA$_{\text{Affine-Diag}}$ | 51.41     | 56.91     | 59.94     |
| HOTDA                       | 47.51     | 47.64     | 62.55     |
| GMM-OTDA$_{\text{T}}$       | **58.35** | 56.57     | 74.10     |
| GMM-OTDA$_{\text{MAP}}$     | 57.36     | **58.77** | **74.44** |

- This table shows that GMM-OTDA __consistently improves performance over different feature extractors__.
- Furthermore, using diagonal covariances (OTDA$\_{\text{Affine-Diag}}$) consistently outperforms the full covariance setting (OTDA$\_{Affine}$) in the context of estimating an affine OT mapping. This finding proves the usefulness of using diagonal covariance matrices in high-dimensional settings.
- More importantly, our method manages to improve the source-only baseline by a margin of 17.74% using ViT-based features.

These results were added in section 5.3 of our main paper (VisDA-C benchmark). We further added t-SNE visualizations of the different mapping techniques.

## References

Christopher Bishop M. Pattern Recognition and Machine Learning. Information Science and Statistics. Springer, New York, NY, 1 edition, August 2006.

---

> ### Author Response · Authors · 2024-12-03
> **New summary of changes**
>
> Following our discussion with reviewer __MxMx__, we followed his suggestion, and added new comparisons with deep methods in the context of the Tennessee Eastman Benchmark (Appendix B.4), where we included results with 4 deep domain adaptation methods,
>
> - DANN (Ganin et al., 2016) uses a domain confusion loss to learn domain invariant features
> - DAN (Ghifary et al., 2014) uses the MMD to learn domain invariant features
> - WDGRL (Shen et al., 2018) uses the dual Wasserstein distance between domain distributions to learn domain invariant features
> - DeepJDOT (Damodaran et al., 2018) uses the primal Wasserstein distance between domain joint distributions (feature-label joint distribution for the source domain, and a proxy for the feature-label joint of the target domain) to learn domain invariant features
>
> The results are summarized in Figure 9 in the manuscript. We further reduce the figure to the following table,
>
> | Method | Average adaptation performance |
> |-----------|----------------------------------------------|
> | Baseline| 48.01|
> |OTDA$\_{\text{EMD}}$| 60.37 |
> |OTDA$\_{\text{Sinkhorn}}$ | 59.39 |
> |OTDA$\_{\text{Affine}}$| 56.76 |
> |HOTDA|57.34|
> |GMM-OTDA$\_{\text{MAP}}$|59.72|
> |GMM-OTDA$\_{\text{T}}$|59.48|
> |DANN|36.11|
> |DAN|51.28|
> |WDGRL|48.21|
> |DeepJDOT|57.88|
>
> ## References
>
> Yaroslav Ganin, Evgeniya Ustinova, Hana Ajakan, Pascal Germain, Hugo Larochelle, François Laviolette, Mario March, and Victor Lempitsky. Domain-adversarial training of neural networks. Journal of machine learning research, 17(59):1–35, 2016.
>
> Muhammad Ghifary, W Bastiaan Kleijn, and Mengjie Zhang. Domain adaptive neural networks for object recognition. In PRICAI 2014: Trends in Artificial Intelligence: 13th Pacific Rim International Conference on Artificial Intelligence, Gold Coast, QLD, Australia, December 1-5, 2014. Proceedings 13, pp. 898–904. Springer, 2014
>
> Jian Shen, Yanru Qu, Weinan Zhang, and Yong Yu. Wasserstein distance guided representation learning for domain adaptation. In Proceedings of the AAAI conference on artificial intelligence, volume 32, 2018
>
> Bharath Bhushan Damodaran, Benjamin Kellenberger, Rémi Flamary, Devis Tuia, and Nicolas Courty. Deepjdot: Deep joint distribution optimal transport for unsupervised domain adaptation. In Proceedings of the European conference on computer vision (ECCV), pp. 447–463, 2018.

---

### Decision · Action_Editor_BuRx · 2024-12-10

**Recommendation:** Accept with minor revision

**Comment:**

The submission meets the bar for both the Claims and Evidence and the Audience criteria. Minor revision requested:

The scope of the claim about outperforming or being competitive with the state-of-the-art is well explained in the Experiments section of the updated manuscript, but the abstract and conclusion remain ambiguous on that distinction. The authors should make sure that this (i.e., the fact that the SOTA claims pertain to shallow domain adaptation approaches) is clearly explained in the abstract and conclusion before publication.

**Audience:**

All reviewers agree in their final recommendation that the paper meets the bar for acceptance in terms of interest to the TMLR audience, and their reviews do not raise any major concern in that respect.

**Claims And Evidence:**

All reviewers agree in their final recommendation that the paper meets the bar for acceptance in terms of claims and evidence. Their reviews highlight the paper's extensive experiments (zrmE) and the proposed approach's solid theoretical grounding (zrmE, MxMx).

Reviewers' concerns over the lack of more complex and widely-used benchmarks (MxMx, hmXt, zrmE) are addressed to their satisfaction by the addition of results on the VisDA benchmark. Some reviewers (MxMx, hmXt) express concerns over the claim that the proposed approach outperforms competing methods given the lack of a comparison against e.g. CoVi or FixBI. The authors clarify that the claim is scoped to shallow domain adaptation approaches. This more restricted scope remains a concern for Reviewer hmXt, but ultimately the claim (scoped to shallow domain adaptation approaches) is well-supported by empirical evidence in the submission.